 TOOLS AND RESOURCES

# Inference of nonlinear receptive field subunits with spike-triggered clustering

**Nishal P Shah**[1]*, **Nora Brackbill**[2], **Colleen Rhoades**[3], **Alexandra Kling**[4,5,6], **Georges Goetz**[4,5,6], **Alan M Litke**[7], **Alexander Sher**[8], **Eero P Simoncelli**[9,10]*, **EJ Chichilnisky**[4,5,6]*

[1]Department of Electrical Engineering, Stanford University, Stanford, United States; [2]Department of Physics, Stanford University, Stanford, United States; [3]Department of Bioengineering, Stanford University, Stanford, United States; [4]Department of Neurosurgery, Stanford School of Medicine, Stanford, United States; [5]Department of Ophthalmology, Stanford University, Stanford, United States; [6]Hansen Experimental Physics Laboratory, Stanford University, Stanford, United States; [7]Institute for Particle Physics, University of California, Santa Cruz, Santa Cruz, United States; [8]Santa Cruz Institute for Particle Physics, University of California, Santa Cruz, Santa Cruz, United States; [9]Center for Neural Science, New York University, New York, United States; [10]Howard Hughes Medical Institute, Chevy Chase, United States

*For correspondence:
bhaishahster@gmail.com (NPS);
eero.simoncelli@nyu.edu (EPS);
ej@stanford.edu (EJC)

**Competing interests:** The authors declare that no competing interests exist.

**Abstract** Responses of sensory neurons are often modeled using a weighted combination of rectified linear subunits. Since these subunits often cannot be measured directly, a flexible method is needed to infer their properties from the responses of downstream neurons. We present a method for maximum likelihood estimation of subunits by soft-clustering spike-triggered stimuli, and demonstrate its effectiveness in visual neurons. For parasol retinal ganglion cells in macaque retina, estimated subunits partitioned the receptive field into compact regions, likely representing aggregated bipolar cell inputs. Joint clustering revealed shared subunits between neighboring cells, producing a parsimonious population model. Closed-loop validation, using stimuli lying in the null space of the linear receptive field, revealed stronger nonlinearities in OFF cells than ON cells. Responses to natural images, jittered to emulate fixational eye movements, were accurately predicted by the subunit model. Finally, the generality of the approach was demonstrated in macaque V1 neurons.

## Introduction

The functional properties of sensory neurons are often probed by presenting stimuli and then inferring the rules by which inputs from presynaptic neurons are combined to produce receptive field selectivity. The most common simplifying assumption in these models is that the combination rule is *linear*, but it is well-known that many aspects of neural response are highly nonlinear. For example, retinal ganglion cells (RGCs) are known to be driven by nonlinear 'subunits' (*Hochstein and Shapley, 1976*), which reflect signals from presynaptic bipolar cells that are rectified at the synapse onto RGCs (*Demb et al., 1999*; *Demb et al., 2001*; *Borghuis et al., 2013*). This subunit computation is fundamentally nonlinear because hyperpolarization of one bipolar cell input does not cancel depolarization of another. The subunit architecture endows RGCs with sensitivity to finer spatial detail than would arise from a linear receptive field (*Hochstein and Shapley, 1976*; *Demb et al., 2001;*; *Baccus et al., 2008*; *Crook et al., 2008*; *Schwartz et al., 2012*) and has been implicated in the processing of visual features like object motion and looming (*Olveczky et al., 2007*; *Münch et al.,*

*2009*). As another example, complex cells in primary visual cortex are thought to perform subunit computations on simple cell inputs, producing invariance to the spatial location of a stimulus. Indeed, subunits appear to be a common computational motif in the brain, and inferring their functional properties from neural recordings requires the development of effective estimation methods.

To this end, simple techniques have been used to reveal the presence and typical sizes of subunits (*Hochstein and Shapley, 1976*), and more elaborate techniques have been used in specific experimental settings (*Emerson et al., 1992*; *Paninski, 2003*; *Sharpee et al., 2004*; *Rust et al., 2005*; *Schwartz et al., 2006*; *Pillow and Simoncelli, 2006*; *Rajan and Bialek, 2013*; *Park et al., 2013*; *Theis et al., 2013*; *Kaardal et al., 2013*; *Freeman et al., 2015*; *Liu et al., 2017*; *Maheswaranathan et al., 2018*; *Shi et al., 2019*). However, no widely accepted general computational tools exist for inferring the structure of nonlinear subunits providing inputs to a neuron, such as their individual sizes, shapes, and spatial arrangement. Such a tool must be robust, efficient, and applicable, possibly with simple modifications, in a variety of different experimental contexts.

Here, we present a novel method for estimating the properties of subunits providing inputs to a recorded neuron. The method was derived as a maximum likelihood procedure for estimating subunit model parameters and took the form of a spike-triggered clustering algorithm. As such, it provided a natural generalization of spike-triggered averaging, which is widely used to estimate linear receptive fields because of its conceptual simplicity, robustness, and straightforward interpretation. In populations of recorded parasol RGCs in macaque retina, the new method revealed a gradual partitioning of receptive fields with a hierarchical organization of spatially localized and regularly spaced subunits, consistent with the input from a mosaic of bipolar cells, and generalized naturally and parsimoniously to a population of cells. Unlike other approaches, the technique optimized a simple and explicit model of neural response, required no assumptions about the spatio-temporal properties of subunits, and was validated on an extensive data set. The structure of the model permitted prior information, such as spatial locality, to be incorporated naturally in cases of limited data. A novel closed-loop 'null' stimulus revealed that the subunit model was substantially more accurate than a linear model in explaining RGC responses, a finding that also extended to natural stimuli. Responses to null stimuli were stronger in OFF parasol cells than ON parasol cells, consistent with their stronger nonlinear responses (*Demb et al., 2001*; *Chichilnisky and Kalmar, 2002*; *Turner and Rieke, 2016*). Application of the estimation method to complex cells in primary visual cortex revealed subunits with expected spatiotemporal structure, demonstrating that the approach generalizes to other experimental contexts and neural circuits.

## Results

The goal is to develop a method to estimate the nonlinear subunits of recorded neurons, examine the spatial and temporal properties of the estimated subunits, and verify their predictions for a wide range of stimuli.

### Subunit response model and parameter estimation

The method is developed as an optimal estimation procedure for a generic subunit model in which responses are described as an alternating cascade of two linear-nonlinear (LN) stages (*Figure 1A*). In the first LN stage, visual inputs over space and recent time are weighted and summed by linear subunit filters, followed by an exponential nonlinearity. In the second LN stage, the subunit outputs are weighted by non-negative scalars, summed, and passed through a final output nonlinearity. This yields the neuron's firing rate, which drives a Poisson spike generator.

To predict neural responses using the model, the model parameters (subunits, weights, and nonlinearity) must be estimated from recorded data. The estimation problem is difficult because the negative log-likelihood is not a convex function of the parameters. An efficient solution is presented with two steps (see Materials and methods): (1) obtain a maximum likelihood estimate of the subunits by identifying clusters in the space of spike-triggered stimuli (*Figure 1B*), (2) estimate the parameters of the output nonlinearity using standard optimization methods. This *spike-triggered clustering* solution offers a simple and robust generalization of the commonly used spike-triggered averaging method for receptive field estimation (see *Chichilnisky, 2001*). Intuitively, clustering is effective because stimuli that produce spikes will typically fall into distinct categories based on which subunit

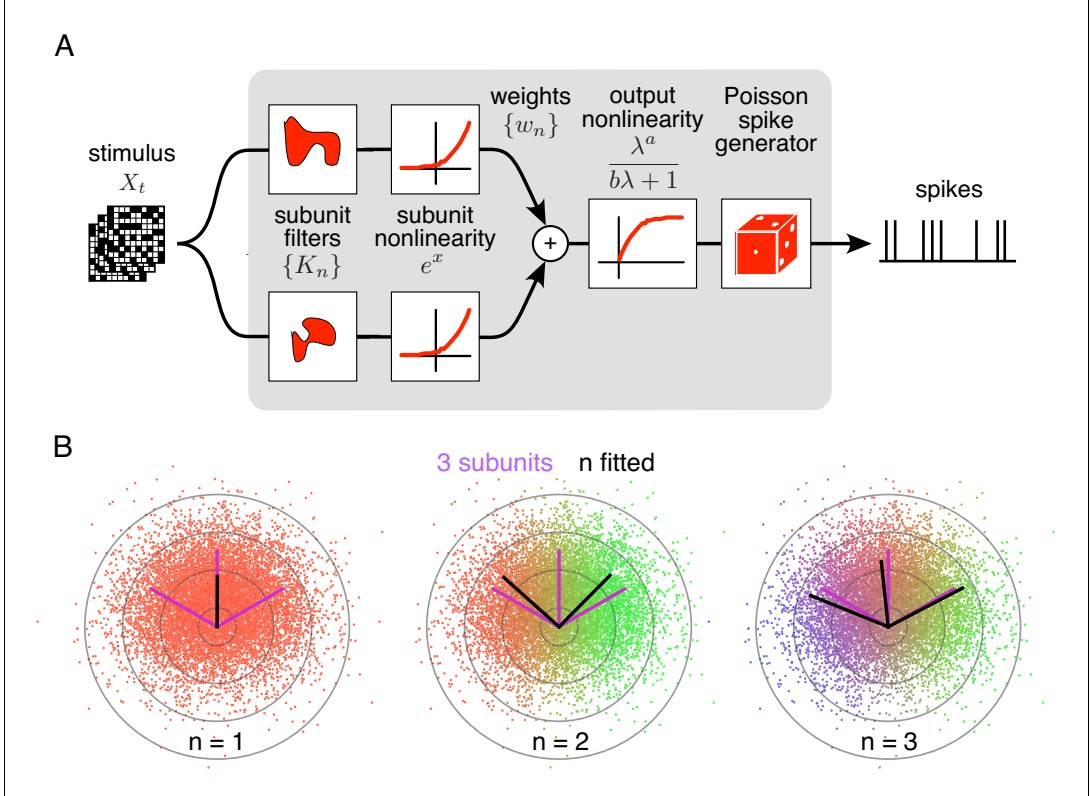

**Figure 1.** Spiking response model, and estimation through spike-triggered stimulus clustering. (**A**) The model is constructed as a cascade of two linear-nonlinear (LN) stages. In the first stage, subunit activations are computed by linearly filtering the stimulus ($X_t$) with kernels ($K_n$) followed by an exponential nonlinearity. In the second stage, a sum of subunit activations, weighted by ($w_n$), is passed through a saturating output nonlinearity (*g*), yielding a firing rate that drives an inhomogeneous Poisson spike generator. (**B**) Estimation of subunits in simulated three-subunit model cell. Responses are generated to two-dimensional Gaussian stimuli (iso-probability circular contours shown) with three subunits (magenta lines), generating a spike-triggered stimulus ensemble (colored dots). Weights for different subunits are assigned to each spike triggered stimulus, with colors indicating the relative weights for different subunits. Subunits are then estimated by weighted summation of spike triggered stimulus ensemble (black lines). Soft-clustering of spike triggered stimuli with different number of clusters results in progressive estimation of the underlying subunits. Clustering with correct number of subunits (right panel) results in estimated subunits (black lines) aligned with the true subunits.

The online version of this article includes the following figure supplement(s) for figure 1:

**Figure supplement 1.** Validation of the subunit fitting algorithm on simulated RGC data.

(s) they activate. Simulations with a RGC model show that this procedure yields accurate estimates of model parameters when sufficient data are available (*Figure 1—figure supplement 1*).

The remaining model parameter, the number of subunits, is discrete and is therefore typically selected by evaluating model predictions on held-out data (i.e. cross-validation) (*Figure 2C*). However, in some cases, it is desirable to either compare the results obtained with different numbers of estimated subunits (in order to study the strongest sources of nonlinearity; *Figure 2A,B*, 4, 5–8), or to use a fixed number of subunits (to compare variants of the algorithm; *Figure 3*).

In what follows, results are presented using the core estimation approach, followed by several modifications appropriate for analyzing data from different experimental conditions. Specifically, consider two situations with data limitations: short recording durations, and biased stimulus statistics. For short durations, the core method is modified to include regularization that assumes localized spatial structure of subunits (see Materials and methods, *Figures 3,4*). For biased stimuli (e.g. natural scenes), subunit filters are estimated using responses from white noise, then the output nonlinearities are fine-tuned (Figures 6,7).

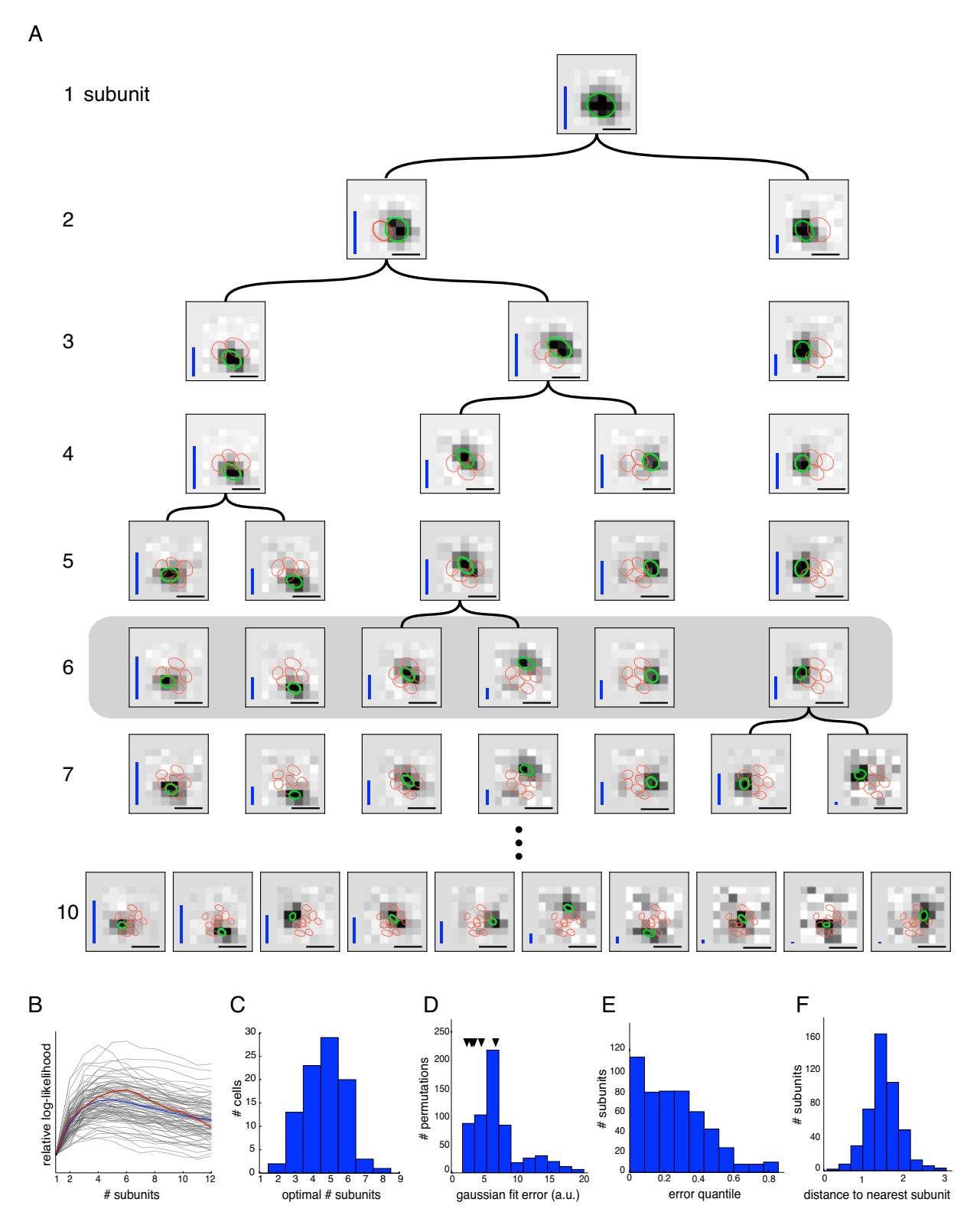

**Figure 2.** Estimated subunit properties. (**A**) Subunits, shown as grayscale images, estimated from OFF parasol cell responses to 24 min of white noise. Each pixel was temporally prefiltered with a kernel derived from the spike-triggered average (STA). Rows show estimated spatial subunits for successive values of N. Subunit locations are indicated with ellipses (green for same subunit, red for other subunits from the same fit), corresponding to the contour of a fitted two-dimensional Gaussian with standard deviation equal to the average nearest neighbor separation between subunits. As N

*Figure 2 continued on next page*

*Figure 2 continued*

increases, each successive set of subunits may be (approximately) described as resulting from splitting one subunit into two (indicated by lines). Large N (e.g. last row) yields some subunits that are noisy or overlap substantially with each other. Height of vertical blue bars indicate the relative strength (average contribution to the cell's firing rate over stimulus ensemble, ignoring the output nonlinearity) of each subunit (see *Equation 6* in Materials and methods). Horizontal black bars indicate spatial scale (150µm). (B) Log-likelihood as a function of number of subunits (relative to single subunit model) for 91 OFF parasol cells (black) on 3 min of held-out test data, averaged across 10 random initializations of the model, from a distinct randomly sampled training data (24 min from remaining 27 min of data). Population average is shown in blue and the example cell from (A) is shown in red. (C) Distribution of optimal number of subunits across different cells, as determined by cross-validated log-likelihood on a held-out test set for OFF parasol cells. (D, E) Spatial locality of OFF parasol subunits, as measured by mean-squared error of 2D gaussian fits to subunits after normalizing with the maximum weight over space. Control subunits are generated by randomly permuting pixel weights for different subunits within a cell. For this analysis, the optimal number of subunits was chosen for each cell. (D) Distribution of MSE values for randomly permuted subunits for the cell shown in (A). MSE of six (optimal N) estimated subunits indicated with black arrows. (E) Distribution of quantiles of estimated OFF parasol subunits, relative to the distribution of MSE values for permuted subunits, across all cells and subunits. Null hypothesis has uniform distribution between 0–1. (F) Distribution of distances to nearest neighboring subunit within each OFF parasol cell. Distances are normalized by geometric mean of standard deviation of the gaussian fits along the line joining center of subunits. For this analysis, each cell is fit with five subunits (most frequent optimum from (C)).

The online version of this article includes the following figure supplement(s) for figure 2:

**Figure supplement 1.** Gradual partitioning of the receptive field into subunits by hierarchical clustering.

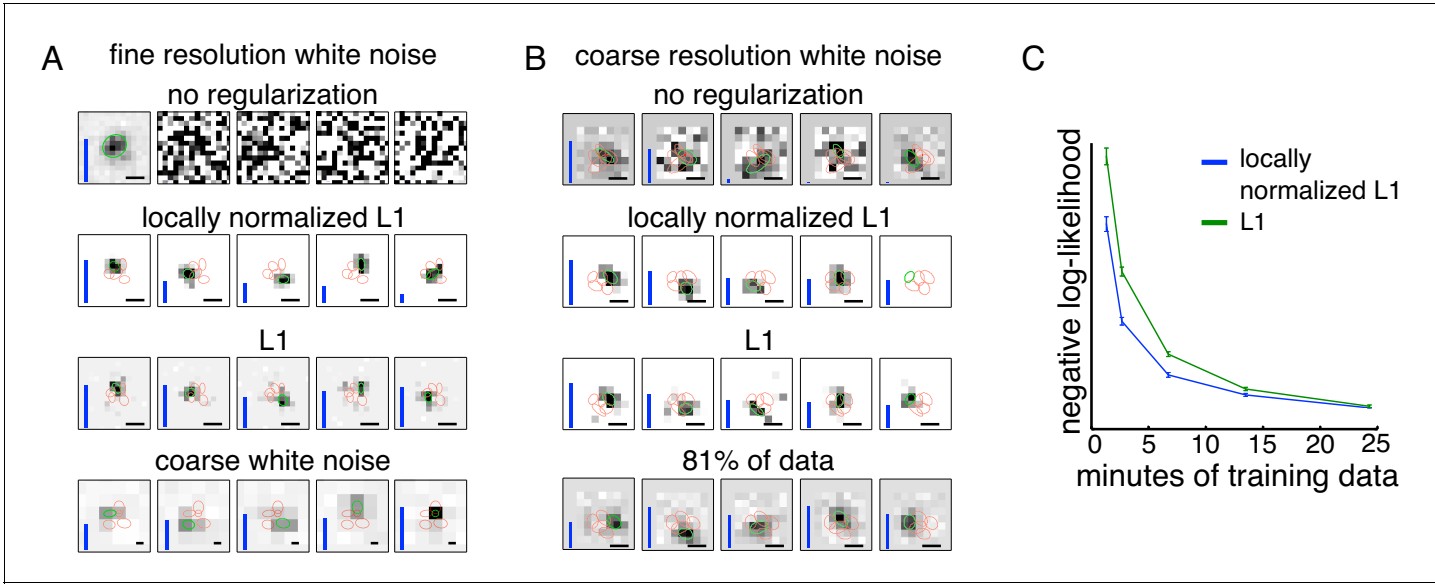

**Figure 3.** Spatially localized subunit estimation. Comparison of different regularizers for estimating subunits using limited data. Examples of OFF parasol cells are shown. (A) Five subunits (most frequent optimum across cells from *Figure 2*) estimated using all data (1 hr 37 min) for fine resolution white noise without regularization (top row). The first estimated subunit is identical to the full receptive field and the others are dominated by noise. Locally normalized L1 (second row) and L1 (third row) regularization both give spatially localized subunits, with L1 regularization leaving more noisy pixels in background. In both cases, optimal regularization strength was chosen (from 0 to 1.8, steps of 0.1) based on performance on held-out validation data. The contours reveal interdigitation of subunits (red lines). Subunits estimated using white noise with 2.5x coarser resolution (24 min) and no regularization are larger, but at similar locations as subunits with fine resolution (bottom row). Height of vertical blue bars indicate the relative strength (average contribution to response) for each subunit (see *Equation 6* in Materials and methods). Scale bar: 75 µm (B) For the cell in *Figures 1A*, 5 subunits estimated using the 3 min (10% of recorded data) of coarse resolution white noise responses are noisy and non-localized (top row). Similar to the fine case, using locally normalized L1 (second row) and L1 (third row) regularization both give spatially localized subunits, with L1 regularization subunits having noisier background pixels. The regularization strength (between 0 and 2.1, steps of 0.1) was chosen to maximize log-likelihood on held out data (last 5 min of data). Subunits estimated using 24 min (81% of data) of data are spatially localized and partition the receptive field (bottom row). Vertical bars same as (A). Scale bar: 150 µm (C) Held out log-likelihood for a 5 subunit model estimated from varying durations of training data with L1 (green) and locally normalized L1 (blue) regularization. Results averaged over 91 OFF parasol cells from *Figure 1A*.

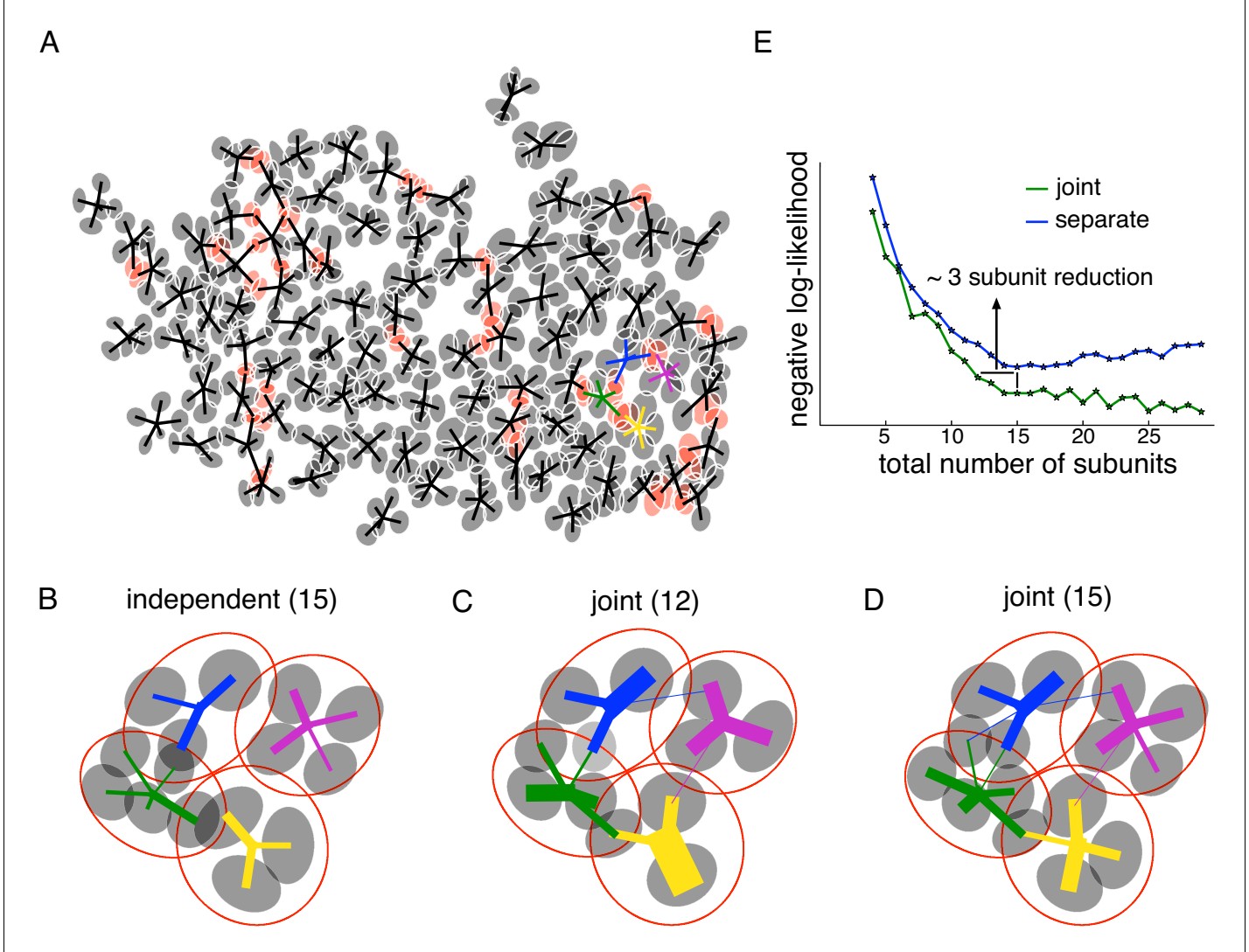

**Figure 4.** Joint estimation of subunits across multiple nearby cells. (**A**) Gaussian fits to the subunits estimated for an entire OFF parasol cell population (5 subunits per cell, with poorly estimated subunits removed). Lines connect center of each cell to its subunits. Subunits which are closer to their nearest neighbor than average (below 15 percentile) are indicated in red. (**B**) Detailed examination of four neighboring cells from the population (color-coded). Number of subunits for each cell is chosen (total 15 subunits) to give the highest summation of log-likelihood (on validation data) for four nearby cells. Gaussian fits to receptive field of the cells (red) and their subunits (gray). Connection strength from a cell (distinct color) to its subunits is indicated by thickness of line. (**C**) A common bank of 12 subunits estimated by jointly fitting responses for all four cells gives similar accuracy as (**B**). Sharing is indicated by subunits connected with lines of different colors (different cells). (**D**) A model with a common bank of 15 subunits (same number as B) gives better performance than estimating the subunits for each cell separately. (**E**) The total negative log-likelihood on test data for these four cells (y-axis) v/s total number of subunits (x-axis) for the two population models with subunits estimated jointly across cells (green) and a combination of separately estimated subunits chosen to maximize total log-likelihood on validation data (blue). Horizontal shift between curves indicates the reduction in total number of subunits by jointly estimating subunits for nearby cells for similar prediction accuracy. The vertical shift indicates better performance by sharing subunits, for a fixed total number of subunits. For both separate and joint fitting, the best value for locally normalized L1 regularization (in 0–3, step size 0.2) for a fixed number of subunits was chosen based on the performance on a separate validation dataset (see Materials and methods).

## Estimated subunits emerge hierarchically and are spatially localized and non-overlapping

To test the subunit model and estimation procedure on primate RGCs, light responses were obtained using large-scale multielectrode recordings from isolated macaque retina (*Chichilnisky and Kalmar, 2002*; *Litke et al., 2004*; *Frechette et al., 2005*). Spiking responses of hundreds of RGCs to a spatiotemporal white noise stimulus were used to classify distinct cell types, by examining the

properties of the spike-triggered average (STA) stimulus (*Frechette et al., 2005*; *Field et al., 2007*). Complete populations of ON and OFF parasol RGCs covering the recorded region were examined.

Fitting a subunit model reliably to these data was aided by decoupling the spatial and temporal properties of subunits. Specifically, although the fitting approach in principle permits estimation of full spatiotemporal subunit filters, the high dimensionality of these filters requires substantial data (and thus, long recordings). The dimension of the parameter space was reduced by assuming that subunit responses are spatio-temporally separable, and that all subunits have the same temporal filter (consistent with known properties of retinal bipolar cells that are thought to form the RGC subunits, see *Enroth-Cugell et al., 1983* and *Cowan et al., 2016*). The common temporal filter was estimated from the STA (see Materials and methods). The stimulus was then convolved with this time course to produce an instantaneous spatial stimulus associated with each spike time. This temporally prefiltered spatial stimulus was then used to fit a model with purely spatial subunits.

The core spike triggered clustering method (no regularization) was applied to study the spatial properties of subunits with increasing numbers of subunits (N). As expected, setting N = 1 yielded a single subunit with a receptive field essentially identical to the STA, i.e. the spatial filter of a LN model. Interestingly, a model with two subunits partitioned this receptive field into spatially localized regions (*Figure 2A*, second row). Fitting the model with additional subunits (*Figure 2A*, subsequent rows) typically caused one of the subunits from the preceding fit to be partitioned further, while other subunits were largely unchanged. In principle this procedure could be repeated many times to capture all subunits in the receptive field. However, the number of model parameters increases with N, and thus the estimation accuracy decreases, with estimated subunits becoming noisier and exhibiting larger overlap (*Figure 2A*, last row). The partitioning observation suggests the possibility of an estimation approach in which the hierarchical introduction of subunits is built in, with the potential for greater efficiency (see Materials and methods). Explicitly incorporating the hierarchical assumption produces a similar collection of subunits at each level (*Figure 2—figure supplement 1*), reinforcing the idea of subunit partitioning as the number of subunits increases. However, for clarity and brevity, the hierarchical estimation approach is not used for the remainder of this paper.

An optimal number of subunits was chosen for each cell to maximize the cross-validated likelihood (i.e. the likelihood measured on a held out test set - *Figure 2B*). The typical optimum for OFF parasol cells in the recorded data was 4–6 subunits (*Figure 2C*). This optimal value is governed by the true number of subunits, the resolution of the stimulus (which determines the dimensionality of the parameter space, see *Figure 3*), and the amount of data (i.e. number of spikes). Since the ON parasol cells had a smaller optimum number of subunits (one or two subunits provided the optimum fit for 48% of ON parasol cells, while this held for only 3% of OFF parasol cells) and exhibited much smaller increases in likelihood of the subunit model compared to a LN model (not shown), subsequent analysis focuses only on the OFF parasol cells. Note, however, that a model with multiple subunits frequently explained ON parasol data more accurately than a model with one subunit (three or more subunits for 52% of cells), implying that ON parasol cells do have spatial nonlinearities.

The estimated subunits were larger and fewer in number than expected from the size and number of bipolar cells providing input to parasol RGCs at the eccentricities recorded (*Jacoby et al., 2000*; *Schwartz and Rieke, 2011*; *Tsukamoto and Omi, 2015*). However, two other features of estimated subunits suggested a relationship to the collection of underlying bipolar cells contributing to the receptive field. First, estimated subunits were compact in space as shown in *Figure 2A*. This was quantified by comparing each subunit with the collection of subunits derived by randomly permuting the filter values at each pixel location across different subunits of the same cell. Spatial locality for estimated and permuted subunits was then evaluated by the mean-squared error (MSE) of 2D Gaussian fits. Compared to the permuted subunits, the estimated subunits had substantially lower MSE (*Figure 2D,E*). Second, the subunits 'tiled' the RGC receptive field, in that they covered it with minimal gaps and modest overlap. This was quantified by noting that on average, the neighboring subunits for a given cell were separated by ~ 1.5 times their width, with relatively little variation (*Figure 2F*).

Given these observations, a natural interpretation of subunits observed with coarse stimulation is that they correspond to aggregates of neighboring bipolar cells. To test this intuition, the algorithm was applied to white noise responses generated from a simulated RGC with a realistic size and number of subunits in the receptive field (*Jacoby et al., 2000*; *Schwartz and Rieke, 2011*). For data simulated using coarse pixelated stimuli matched to those used in the experiments, the estimated

subunits were fewer and larger than the bipolar cells, with each subunit representing an aggregate of several bipolar cells (*Figure 1—figure supplement 1*). The recovered subunits also exhibited spatial locality and tiling, as in the data.

## Regularization for spatially localized subunit estimation

To estimate subunits with a resolution approaching that of the underlying bipolar cell lattice would require much higher resolution stimuli. But higher resolution stimuli typically require more data. Specifically, to obtain a constant quality estimate of subunits as the stimulus resolution/dimensionality is increased requires a proportional increase in the number of spikes (e.g. see *Dayan and Abbott, 2001*). Furthermore, higher resolution stimuli typically produce lower firing rates, because the effective stimulus contrast is lower. To illustrate the problem, the impact of increased resolution and smaller duration of data were examined separately. Higher resolution stimuli led to worse subunit estimates, even when all the recorded data (97 min) were used (*Figure 3A*, top row). Specifically, of the five estimated subunits, one resembled the receptive field, and the others were apparently dominated by noise and had low weights; thus, the algorithm effectively estimated a LN model. For coarse resolution stimuli, using only 10% of the recorded data led to noisy subunit estimates (*Figure 3B*, top row) compared to the estimates obtained with 81% of the data (24 min, *Figure 3B*, last row). These observations suggest that it would be useful to modify the core clustering algorithm by incorporating prior knowledge about subunit structure, and thus potentially obtain more accurate estimation with limited data.

To accomplish this, a regularizer was introduced to encourage the spatial locality of estimated subunits (*Figure 2*). Previous studies (*Liu et al., 2017*; *Maheswaranathan et al., 2018*; *McFarland et al., 2013*) have used L1 regularization (a common means of inducing sparsity), but L1 is agnostic to spatial locality, and indeed is invariant to spatial rearrangement. Thus, a novel locally normalized L1 (LNL1) regularizer was developed that penalizes large weights, but only if all of the neighboring weights are small: $L(w_i) = \frac{|w_i|}{\epsilon + \sum_{j \in \text{neighbor}(i)} |w_j|}$ for $\epsilon = 0.01$, a value smaller than typical nonzero weights which are on the order of unity. This penalty is designed to have a relatively modest influence on subunit size compared to L1, which induces a strong preference for smaller subunits. The LNL1 penalty was incorporated into the estimation procedure by introducing a proximal operator in the clustering loop (see Materials and methods). In this case, the appropriate projection operator is an extension of the soft-thresholding operator for the L1 regularizer, where the threshold for each pixel depends on the strength of neighboring pixels. The regularization strength is chosen to maximize cross-validated likelihood (see Materials and methods).

Introducing the LNL1 prior improved subunit estimates for both of the limited data situations (high resolution, or small duration) presented above. Since the optimal number of subunits can vary for different regularization methods, the subunits estimated from different regularization methods were compared after fixing the total number of subunits to the most likely optimum across cells (*Figure 2*). In the case of the coarse resolution data, both L1 and LNL1 priors produced spatially localized subunits (*Figure 3B*, middle rows) whose locations matched the location of the subunits estimated using more data without regularization (*Figure 3B*, last row). This suggests that the proposed prior leads to efficient estimates without introducing significant biases. Relative to L1 regularization, LNL1 regularization yielded fewer apparent noise pixels in the background (*Figure 3B*, middle rows) and a larger optimum number of subunits (not shown). This improvement in spatial locality is reflected in more accurate response prediction with LNL1 regularization (*Figure 3C*).

In the case of high dimensional stimuli (*Figure 3A*, top row), both L1 and LNL1 priors were successful in estimating compact subunits (*Figure 3A*, middle rows) and matched the location of subunits obtained with a coarser stimulus (*Figure 3A*, bottom row). The estimated subunits tiled the receptive field, as would be expected from an underlying mosaic of bipolar cells. The LNL1 prior also led to subunit estimates with fewer spurious pixels compared to the L1 prior. Note that although these spurious pixels can also be suppressed by using a larger weight on the L1 prior, this also yielded smaller subunit size, which produced less accurate response predictions, while LNL1 had a much smaller effect on estimated subunit size (not shown). Hence, in both the conditions of limited data examined, the novel LNL1 prior yielded spatially localized subunits, fewer spurious pixels, and a net improvement in response prediction accuracy.

## Parsimonious modeling of population responses using shared subunits

To fully understand visual processing in retina it is necessary to understand how a population of RGCs coordinate to encode the visual stimulus in their joint activity. The core spike triggered clustering method extends naturally to joint identification of subunits in populations of neurons. Since neighboring OFF parasol cells have partially overlapping receptive fields (*Gauthier et al., 2009*), and sample a mosaic of bipolar cells, neighboring cells would be expected to receive input from common subunits. Indeed, in mosaics (lattices) of recorded OFF parasol cells, estimated subunits of different cells were frequently closer than the typical nearest-neighbor subunit distances obtained from single-cell data. For example, in one data set, the fraction of subunits from different cells closer than 1SD of a Gaussian fit was 14%, substantially higher than the predicted value of 9% based on the separation of subunits within a cell (*Figure 2F*). Examination of the mosaic suggests that these closely-spaced subunits correspond to single subunits shared by neighboring RGCs (*Figure 4A*), that is, that they reflect a shared biological substrate. If so, then joint estimation of subunits from the pooled data of neighboring RGCs may more correctly reveal their collective spatial structure.

This hypothesis was tested by fitting a common 'bank' of subunits to simultaneously predict the responses for multiple nearby cells. Specifically, the firing rate for the $i^{\text{th}}$ neuron was modeled as: $R_i = g_i(\sum_n w_{n,i} exp(K_n \cdot X_t))$. Here $X_t$ is the stimulus, $K_n$ are the filters for the common population of subunits, $g_i$ is the output nonlinearity for the $i^{\text{th}}$ neuron, and $w_{n,i}$ is the non-negative weight of the $n^{\text{th}}$ subunit to the $i^{\text{th}}$ neuron, which is 0 if they are not connected. The model parameters were estimated by maximizing the sum of log-likelihood across cells, again in two steps. For the first step, the output nonlinearities were ignored for each cell and a common set of subunit filters was found that clustered the spike-triggered stimuli for all the cells simultaneously. Specifically, on each iteration, a) the relative weights for different subunits were computed for each spike-triggered stimulus and b) the cluster centers were updated using the weighted sum of spike-triggered stimuli across different cells (see Materials and methods). Since the subunit filters could span the receptive fields of all the cells, leading to higher dimensionality, the spatial locality (LNL1) prior was used to efficiently estimate the subunits (*Figure 3*). In the second step of the algorithm, the non-linearities were then estimated for each cell independently (similar to *Figure 1*).

To quantify the advantages of joint subunit estimation, the method was applied to a group of four nearby OFF parasol cells. As a baseline, models with different numbers of subunits were estimated for each cell separately, and the combination of 15 subunits that maximized the cross-validated log-likelihood when summed over all four cells was chosen (*Figure 4B*). Then, a joint model with 12 subunits was fitted. Similar test log-likelihood was observed using the joint model, which effectively reduced model complexity by three subunits (*Figure 4E*, horizontal line). Examination of the subunit locations revealed that pairs of overlapping subunits from neighboring cells in separate fits were replaced with one shared subunit in the joint model (*Figure 4C*). Also, a joint model with 15 subunits showed a higher prediction accuracy than a model in which the 15 subunits were estimated separately (*Figure 4E*, vertical line). In this case, sharing of subunits leads to larger number of subunits per cell (*Figure 4D*).

This trend was studied for different number of subunits, by comparing the total test negative log-likelihood associated with the joint model (*Figure 4E*, green) and the best combination of subunits from separate fits to each cell based on validation data (*Figure 4E*, blue). Joint fitting provided more accurate predictions than separate fits, when using a large total number of subunits (>12). For a smaller number of subunits, a horizontal shift in the observed likelihood curves revealed the reduction in the number of subunits that was obtained with joint estimation, without a reduction in prediction accuracy. In sum, jointly estimating a common collection of subunits for nearby cells resulted in a more parsimonious explanation of population responses.

## Subunit model explains spatial nonlinearities revealed by null stimulus

To more directly expose the advantage of a subunit model over commonly used LN models of RGC light response, experiments were performed using a visual stimulus that elicits no response in a LN model (or other model that begins with linear integration of the visual stimulus over space, for example *Pillow et al., 2008*). An example of such a stimulus is a contrast-reversing grating stimulus centered on the RF (*Hochstein and Shapley, 1976*; *Demb et al., 1999*). However, to avoid recruiting additional nonlinearities, a stimulus was developed that has spatio-temporal statistics very

similar to the high-entropy white noise stimulus used for model characterization. This 'null' stimulus was obtained by manipulating white noise so as to eliminate responses of the linear RF.

Specifically, the computation performed by a hypothetical LN neuron can be represented in a stimulus space in which each axis represents stimulus contrast at a single pixel location (*Figure 5A*). In this representation, the LN response is determined by the projection of the stimulus vector onto the direction associated with the weights in the neuron's spatial RF. Thus, a null stimulus orthogonal to the RF can be computed by subtracting a vector with the same RF projection from the original stimulus (*Figure 5B*). In addition to the orthogonality constraint, two other constraints are imposed in constructing the stimulus: (i) the range of pixel values is bounded for the stimulus display, and (ii) the variance of each pixel over time is held constant (as in white noise), to prevent gain changes in photoreceptors (see Materials and methods). Given these constraints, the resulting stimulus is the *closest* stimulus to the original white noise stimulus that yields zero response in a LN neuron. Note that although this approach is described purely in the spatial domain, it generalizes to the full spatio-temporal RF, and the procedure based on the spatial RF alone will produce a null stimulus for a spatio-temporal linear filter that is space-time separable (see Materials and methods).

The ability to silence RGC responses with null stimuli was tested as follows. A 10 s movie of white noise was projected into the intersection of the spatial null spaces corresponding to all cells in a region, of a single type (e.g. OFF parasol). Responses were recorded to 30 repeated presentations of both the white noise and the null stimulus. RGC firing rates showed a substantial modulation for both white noise and the null stimulus that was highly reproducible across repeats (*Figure 5C,D*), inconsistent with linear integration by the RGC over its RF. Note that the null stimulus modulated OFF parasol cells more strongly than ON parasol cells, consistent with previous results obtained with white noise and natural scene stimuli (*Demb et al., 2001*; *Chichilnisky and Kalmar, 2002*; *Turner and Rieke, 2016*); henceforth, analysis is focused on OFF parasol cells.

The subunit model substantially captured response nonlinearities revealed by the null stimulus. Because the biased statistics of the null stimulus induce errors in the estimated subunits, subunit filters were first estimated from responses to white noise (step 1 of model estimation, without regularization, *Figure 1*), and response nonlinearities were then estimated from responses to the null stimulus (step 2 of model estimation, *Figure 1*). As expected, the single subunit (LN) model failed to capture responses to the held-out null stimulus (*Figure 6A*, second row). As the number of subunits was increased, model performance progressively increased (*Figure 6A*, subsequent rows). Prediction accuracy, defined as the correlation between the recorded and predicted firing rate, was evaluated across entire populations of OFF parasol cells in three recordings (*Figure 6B*). The single subunit model failed, with prediction accuracy near 0 (mean accuracy : 0.02, 0.12, 0.03 for three retinas). The prediction accuracy gradually increased with the addition of more subunits, saturating at roughly 4–6 subunits (*Figure 6B*, red). In contrast, the addition of subunits showed marginal improvements for white noise stimuli of the same duration (*Figure 6B*, black), despite being measurable with longer stimuli (*Figure 2B*). Thus, the response nonlinearities captured by the subunit model are more efficiently exposed by the null stimulus.

## Subunits enhance response prediction for naturalistic stimuli

A fuller understanding of the biological role of spatial nonlinearities requires examining how they influence responses elicited by natural stimuli. For this purpose, segments of images from the Van Hateren database were used, with a new image presented every second to emulate saccadic eye movements. Between simulated saccades, the image was jittered according to the eye movement trajectories recorded during fixation by awake macaque monkeys (ZM Hafed and RJ Krauzlis, personal communication, *van Hateren and van der Schaaf, 1998*; *Heitman et al., 2016*). Because the biased statistics of natural stimuli make it infeasible to directly estimate subunits, subunit filters and time courses were first estimated using white noise stimuli (1st step, without regularization *Figure 1*). Then, the non-linearities and subunit magnitudes were fitted using natural stimuli (2nd step, *Figure 1*) (see Materials and methods, similar to above). Subsequently, the ability of the subunit model to capture responses to natural stimuli was measured.

Responses to natural stimuli showed a distinct structure, with strong periods of firing or silence immediately following saccades and less frequent firing elicited by image jitter (*Heitman et al., 2016*, *Figure 7A* black raster). This structure was progressively better captured by the model as the number of subunits increased, as shown by the rasters for an OFF parasol cell in *Figure 7A*. This

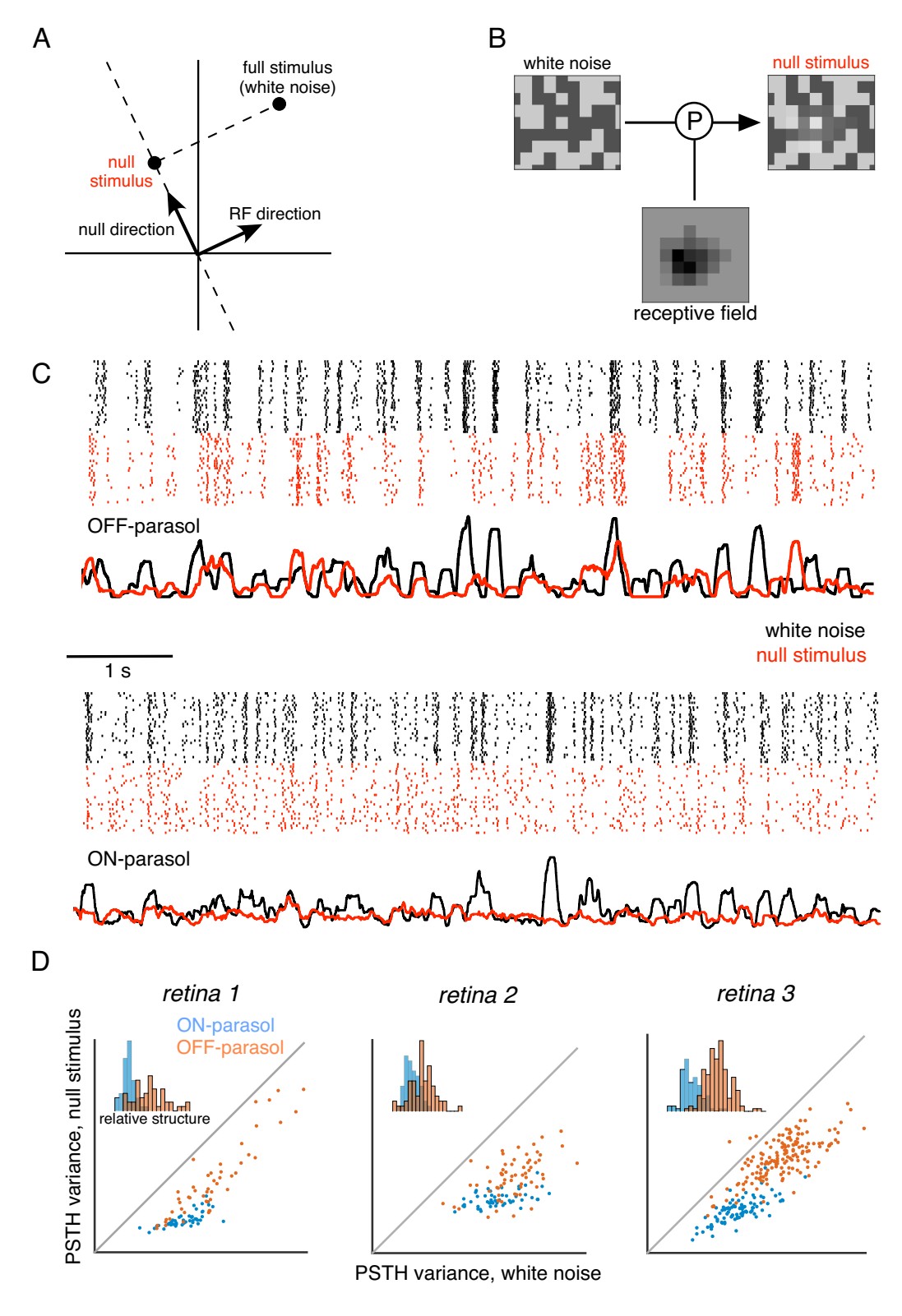

**Figure 5.** Cells respond to stimulus in null space of receptive field. (**A**) Construction of null stimulus, depicted in a two-dimensional stimulus space. Each dimension of stimulus space consists of intensity along a particular pixel. A stimulus frame is represented as a point in this space. Each stimulus frame can be represented geometrically as the sum of the component along the receptive field (RF) direction (the response component of a linear nonlinear

*Figure 5 continued on next page*

*Figure 5 continued*

model) and the component orthogonal to the RF direction (the null component). (B) The null stimulus is constructed by projecting out the RF contribution from each frame of white noise stimulus. (C) Rasters representing responses for an OFF parasol (top) and ON parasol (bottom) cell to 30 repeats of 10 s long white noise (red) and the corresponding null stimulus (black). (D) Response structure for ON parasol (blue) and OFF parasol (orange) populations for white noise (x-axis) and null stimulus (y-axis) across three preparations (different plots). The response structure was measured as variance of PSTH over time. Variance of PSTH converged with increasing number of trials (not shown), suggesting minimal contribution of inter-trial variability to response structure. Insets: Histogram of relative structure in white noise responses that is preserved in null stimulus for ON parasol (blue) and OFF parasol (orange). Relative structure is measured by ratio of response structure in the null stimulus and response structure in white noise stimulus.

finding was replicated in OFF parasol populations in three retinas, and quantified by the correlation between the recorded and predicted PSTH (*Figure 7B*). However, some of the firing events could not be captured even with a large number of subunits, suggesting that models with additional mechanisms such as gain control (*Heeger, 1992*; *Carandini and Heeger, 2012*) or nonlinear interactions between subunits (*Kuo et al., 2016*; *Manookin et al., 2018*) may be required to describe natural scene responses more completely.

Interestingly, further examination revealed that addition of subunits produced greater prediction improvements for responses elicited by image jitter compared to those elicited immediately following simulated saccades (< 250 ms) (*Figure 7B*). This could be explained by the fact that changes in

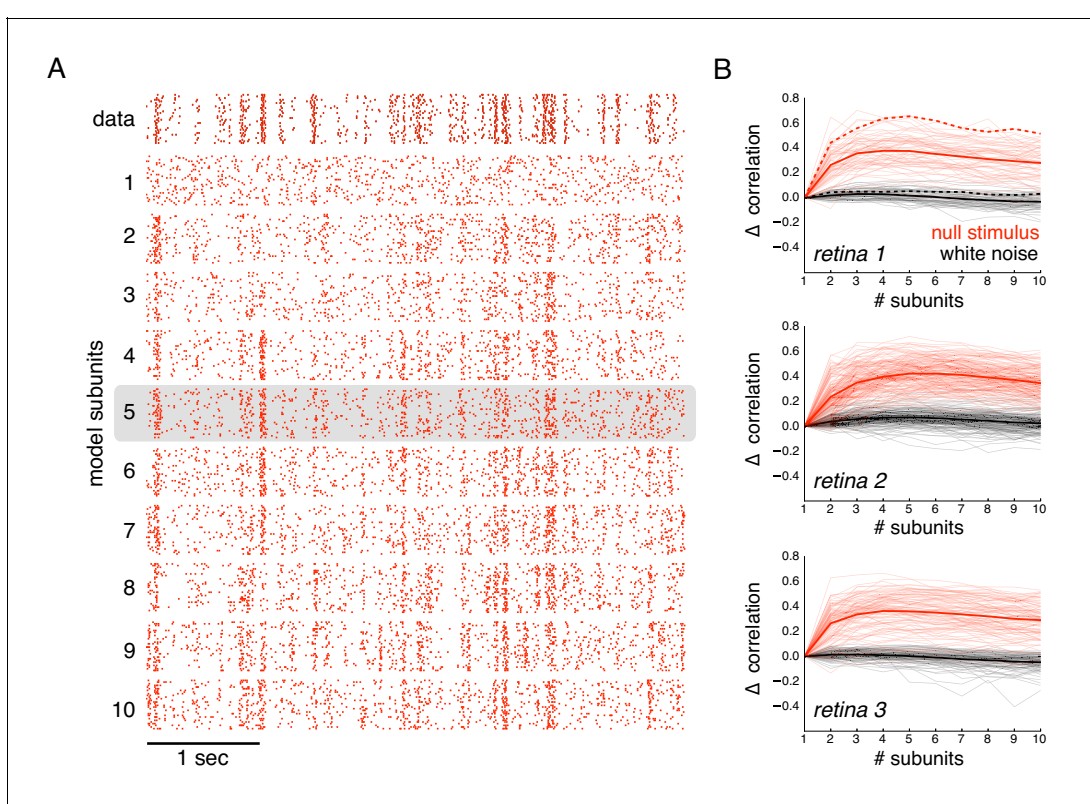

**Figure 6.** Subunits improve prediction of responses to null stimuli. (A) Rasters for recorded responses of an OFF parasol cell to 30 presentations of a 5 s long null stimulus (top row). Predictions of models with increasing (1 to 10) number of subunits (subsequent rows). (B) The change in correlation (relative to the correlation obtained with one subunit) between PSTH for recorded and predicted responses with different numbers of subunits across three preparations. Spatial kernels were estimated from 24 min of non-repeated white noise stimulus, with scales and output nonlinearity estimated from the first 5 s of the repeated stimulus. Performance on the last 5 s of the repeated stimulus was averaged over 10 fits, each with a random subsample of the non-repeated white noise stimulus. Individual OFF parasol cells (thin lines) and population average (thick lines) for the null stimulus (red) and white noise (black). The cell in (A) is indicated with dashed lines.

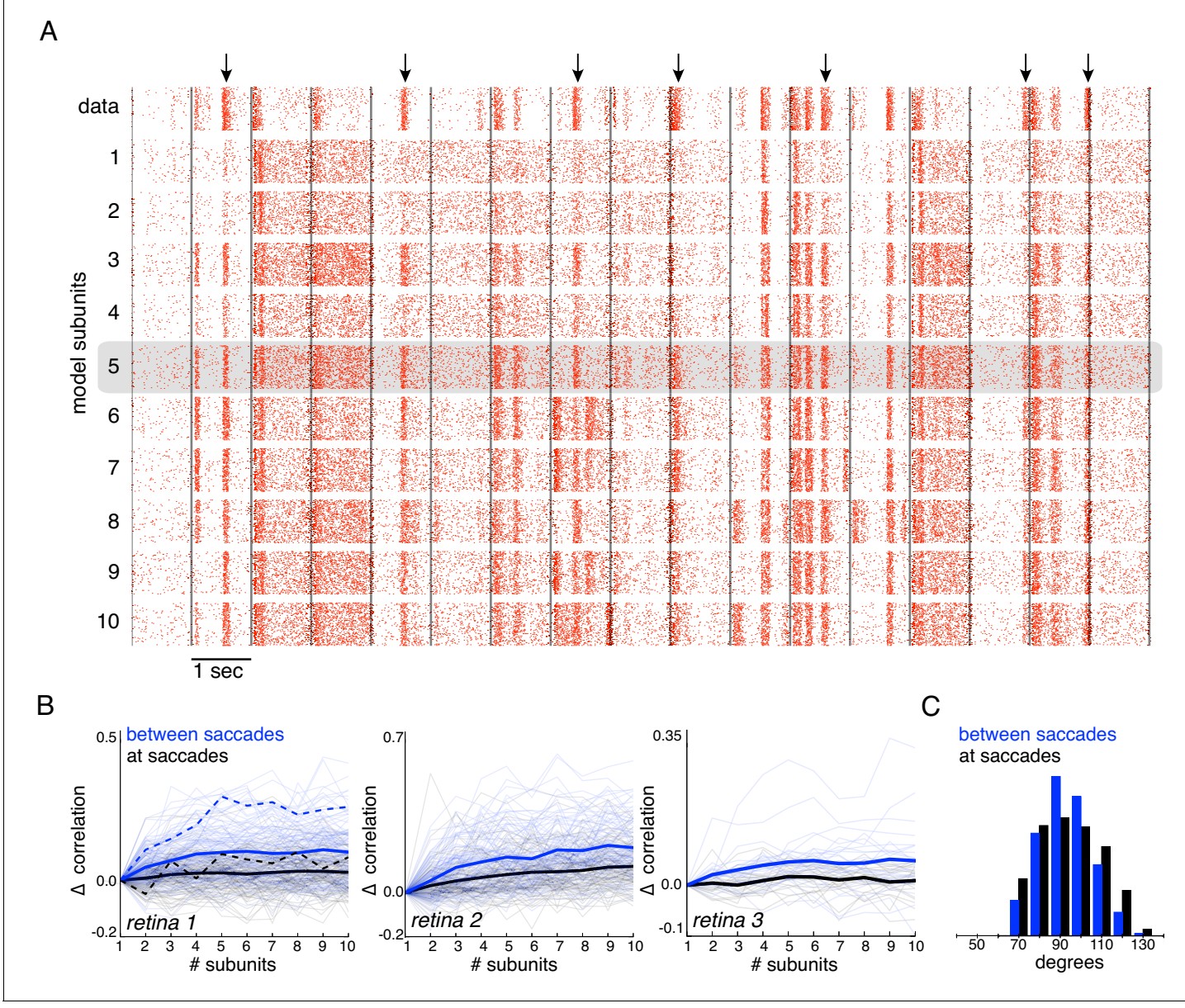

**Figure 7.** Subunits improve response prediction accuracy for naturalistic stimuli. (**A**) Top row: Rasters of responses for an OFF parasol cell from 40 presentations for 30 s long naturalistic stimuli. Natural scene images are presented and spatially jittered, with a new image every 1 s (black lines). Subsequent rows indicate model predictions using different number of subunits (1 to 10), respectively. (**B**) Change in correlation (relative to the correlation obtained with one subunit) between PSTH for recorded and predicted responses with different number of subunits across three preparations. Individual OFF parasol cells (thin lines) and population average (thick lines) for two conditions: 250 ms immediately following saccades (black) and inter-saccadic stimuli (red). Cell from (**A**) shown with dashed lines. (**C**) Distribution of angle between vectors representing the spatial receptive field and natural stimuli (temporally filtered using time course from white noise STA) at saccades (black) and between saccades (blue). Angles with inter-saccadic stimulus are more peaked around 90 degrees, with a standard deviation of 12 degrees, compared with 15 degrees for the saccadic stimulus.

luminance are usually more spatially uniform at saccades, resulting in more linear signaling by RGCs. Indeed, after temporally filtering the visual stimulus, there were fewer stimulus samples in the null space of the receptive field (90 degrees) during saccades than between saccades (*Figure 7C*), and the subunit model provided a more accurate explanation of responses to stimuli in the null space (*Figure 6*). Thus, these results indicate that spatial non-linearities contribute to explaining responses to natural stimuli, especially those elicited by jittering images.

## Application to simple and complex cells in primary visual cortex

Identification of subunits using spike-triggered stimulus clustering ideally would generalize to other neural systems with similar cascades of linear and non-linear operations. As a test of this generality, spike triggered clustering (*Figure 1*) was applied to obtain estimates of space-time subunit filters from responses of V1 simple and complex cells to flickering bars (*Rust et al., 2005*). The estimated subunits showed oriented spatio-temporal structure similar to that observed with other methods that imposed more restrictions (*Rust et al., 2005*; *Vintch et al., 2015*) (because of this similarity, additional priors on the filters were not explored). Increasing the number of subunits typically caused one of the subunits from the preceding fits to be partitioned into two, while other subunits were largely unchanged (*Figure 8*); that is, a hierarchical structure similar to that observed in RGCs (*Figure 2*). The number of cross-validated subunits for the complex cell (N = 8) was greater than for the simple cell (N = 4), consistent with previous work (*Rust et al., 2005*). All subunits for a given cell had similar structure in space and time, but were approximately translated relative to each other, consistent with the similar frequency spectrum (*Figure 8A,B*, bottom row). For the complex cell, this is consistent with the hypothesis that it receives inputs from multiple simple cells at different locations (*Hubel and Wiesel, 1962*; *Adelson and Bergen, 1985*; *Vintch et al., 2015*). Thus, the method

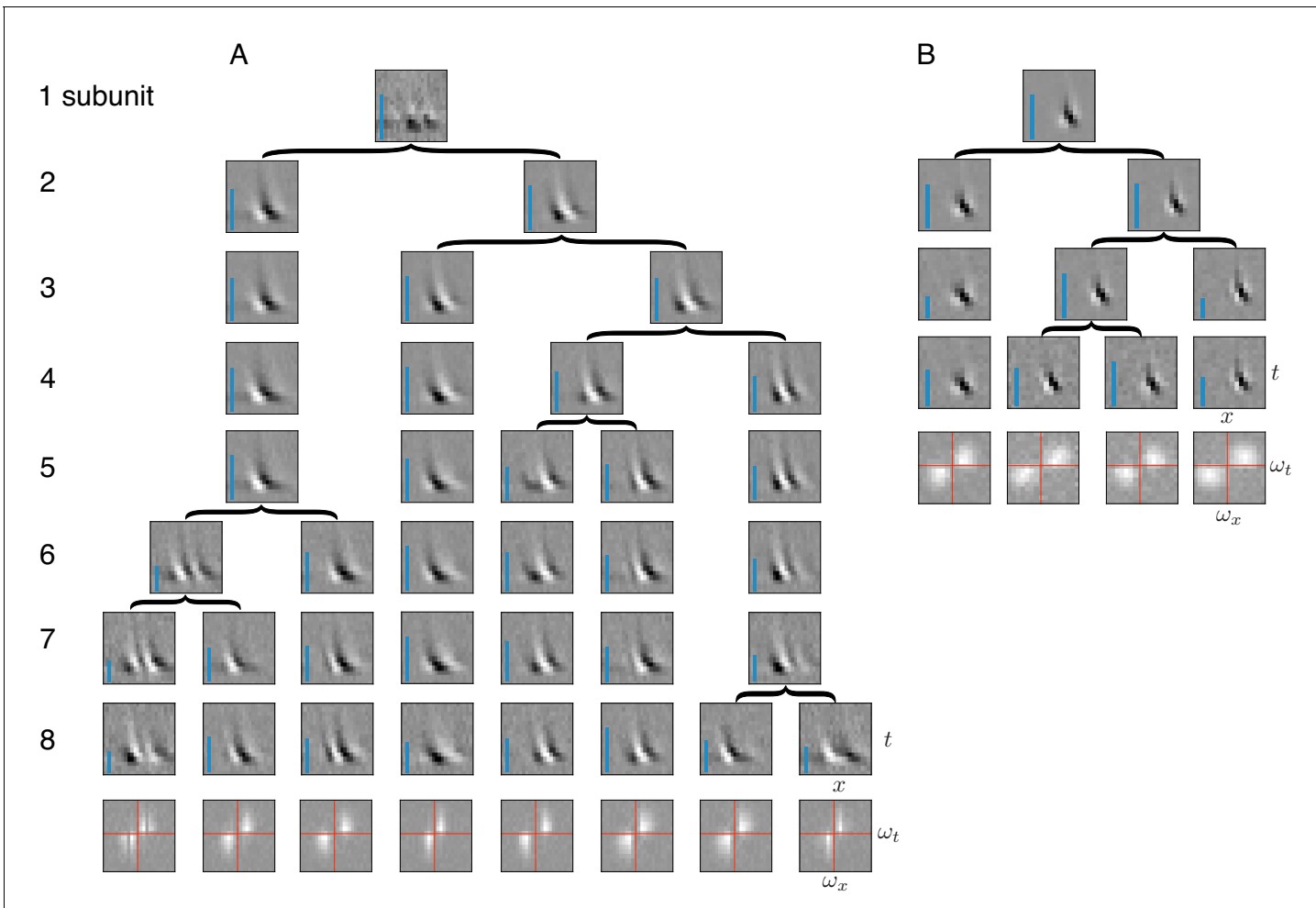

**Figure 8.** Application of subunit model to V1. (A) Hierarchical relationship between subunits estimated using responses to flickering bar stimuli for the complex cell featured in *Rust et al. (2005)*. Rows show estimated spatio-temporal filters for different number of subunits (N). As N increases, each successive set of subunits may be (approximately) described as resulting from splitting one subunit into two (indicated by arrows, see Materials and methods). Largest N represents the optimal number of subunits, determined by cross-validation. Fourier magnitude spectrum of the optimal subunits (bottom row) show translational invariance of estimated subunits. Inset : Relative contribution to cell's firing rate for each subunit indicated by height of blue bar (see *Equation 6* in Materials and methods). (B) Same as (A) for the simple cell from *Rust et al. (2005)*.

presented here extends to V1 data, producing subunit estimates that are not constrained by assumptions of orthogonality (*Rust et al., 2005*) or convolutional structure (*Vintch et al., 2015*), yet broadly resembling the results of previous work (see Discussion and Figure 10 for a detailed comparison).

## Comparison to other methods

Comparisons with recent studies reveal advantages of the spike-triggered clustering approach in terms of efficiency of estimation, and interpretation of estimated subunits. A relevant comparison is the recently published subspace-based approach, spike-triggered non-negative matrix factorization (SNMF) (*Liu et al., 2017*; *Jia et al., 2018*). This method was applied to salamander RGCs, and estimated subunits were shown to be well-matched to independent physiological measurements of bipolar cell receptive fields, an impressive validation. This approach works by collecting the spike triggered stimuli in a matrix and then factorizing the matrix. Because this method differs in a variety of ways from the spike-triggered clustering approach, only the core aspects of the two methods were compared (see Discussion). Both methods were applied without any regularization, and were run without perturbing the parameters after convergence (contrary to the suggestion in *Liu et al., 2017*). The number of subunits used was equal to the true number of subunits in the simulated cell. In these conditions, recovery of subunits revealed two trends. First, for a realistic amount of data, the SNMF approach often produced mixtures of underlying subunits, while the spike-triggered clustering approach produced unique and separate subunits (see *Figure 9*). Second, the spatial structure of extracted subunit was substantially more variable using SNMF across different initializations. These results indicate that, at least in these simulated conditions, the spike-triggered clustering method produces more consistent and accurate estimates of subunit spatial structure than SNMF. Note, however, that choices of initialization and regularization can alter the efficiency of both procedures. A summary of other differences with SNMF is presented in Discussion.

Subunit behaviors have also been captured using convolutional models that rely on the simplifying assumption that all subunit filters are spatially shifted copies of one another (*Eickenberg et al., 2012*; *Vintch et al., 2015*; *Wu et al., 2015*). To perform the comparison, RGC responses were analyzed with a modification of the spike-triggered clustering approach that enforces convolutional subunit structure, with model parameters fitted by gradient descent. The method was applied with locally normalized L1 regularization, shown above to be effective for retinal data (*Figure 10A,B*). As expected, the convolutional model was substantially more efficient, exhibiting lower prediction error for limited training data. However, for longer durations of training data (within the range of typical experiments), the unconstrained model exhibited lower error (*Figure 10A*). The spatial location and resulting layout of subunits obtained from the two methods were similar (*Figure 10B*). However, spike-triggered clustering captured the deviations from a purely convolutional structure, leading to improved performance. Similar results were obtained using spike-triggered clustering without LNL1 regularization (not shown). Similar results were also obtained using data from a V1 complex cell (*Figure 10C,D*). Note that the nearly identical power spectra (*Figure 8*) of the subunits estimated using spike-triggered clustering suggested nearly convolutional structure, without explicitly enforcing this constraint. Note that further improvement in both approaches could potentially be achieved with modifications. For example in the V1 neurons, replacing the exponential nonlinearity with a quadratic nonlinearity (as in *Vintch et al., 2015*) provided more accurate fits for both models, but the performance of spike-triggered clustering was still superior when more data were available (*Figure 10—figure supplement 1*).

## Discussion

We developed an efficient modeling and estimation approach to study a ubiquitous motif of neural computation in sensory circuits: summation of responses of rectified subunits. The approach allowed efficient subunit estimation by clustering of spike-triggered stimuli, and the inclusion of spatial localization regularizers to additionally constrain the fitting under conditions of limited data. The model and fitting procedure extended naturally to populations of cells, allowing joint fitting of shared subunits. The effectiveness of the method was demonstrated by its ability to capture subunit computations in macaque parasol RGCs, specifically in stimulus conditions in which linear models completely

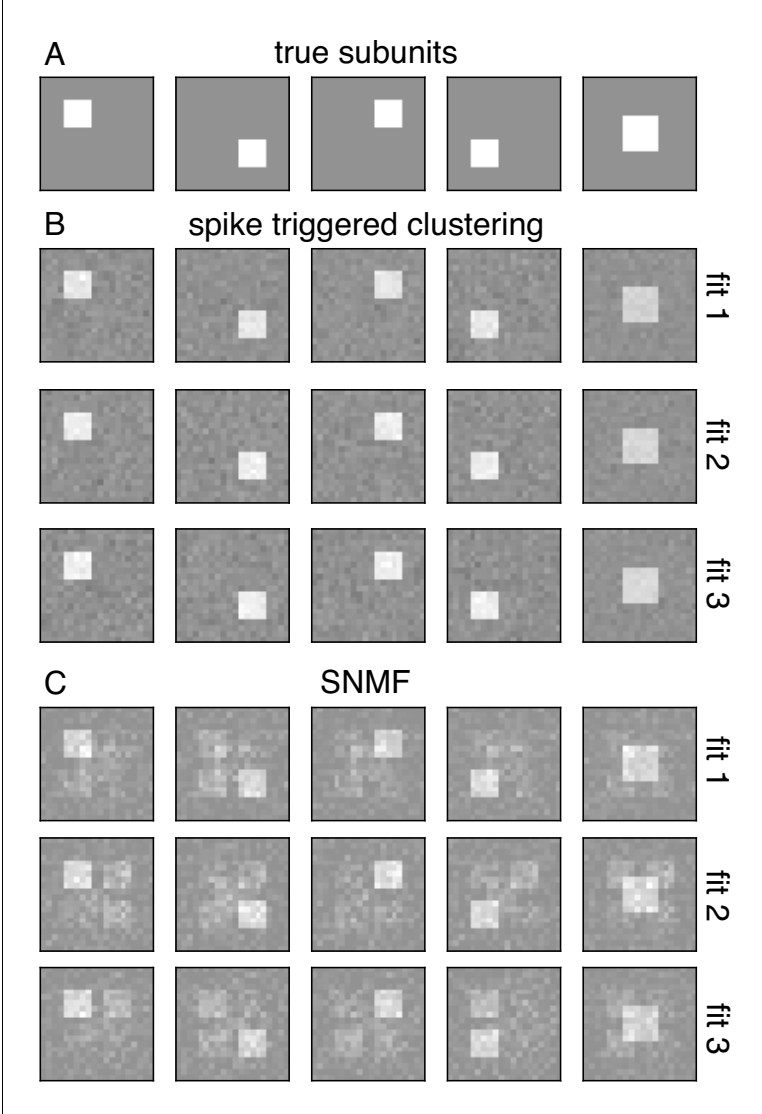

**Figure 9.** Comparison with spike-triggered non-negative matrix factorization. (A) Spatial filters for five simulated subunits. Responses are generated by linearly projecting white noise stimulus onto these spatial filters, followed by exponential subunit nonlinearity, summation across subunits and saturating output nonlinearity. (B) Results from the core spike triggered clustering method (without regularization), applied with N = 5 subunits. Rows indicate fits with different initializations. (C) Results from spike-triggered non-negative matrix factorization (SNMF) applied to simulated data. Code provided by *Liu et al. (2017)*. For comparison with to the present method, SNMF applied with no regularization, N = 5 subunits and one perturbation. Similar to (B), rows correspond to fits with different initializations.

fail, as well in V1 neurons. Below, we discuss the properties of the approach compared to others, and the implications of the application to retinal data.

## Modeling linear-nonlinear cascades

The subunit model is a natural generalization of linear-nonlinear (LN) cascade models, and the estimation procedure is a natural generalization of the spike-triggered estimation procedures that facilitated the widespread use of LN models. In the present model framework, a maximum-likelihood estimate of model parameters leads to a spike-triggered clustering algorithm for obtaining the early linear filters that represent model subunits. These linear components are the key, high-dimensional entities that define stimulus selectivity, and, in the absence of a robust computational framework, would be difficult to estimate.

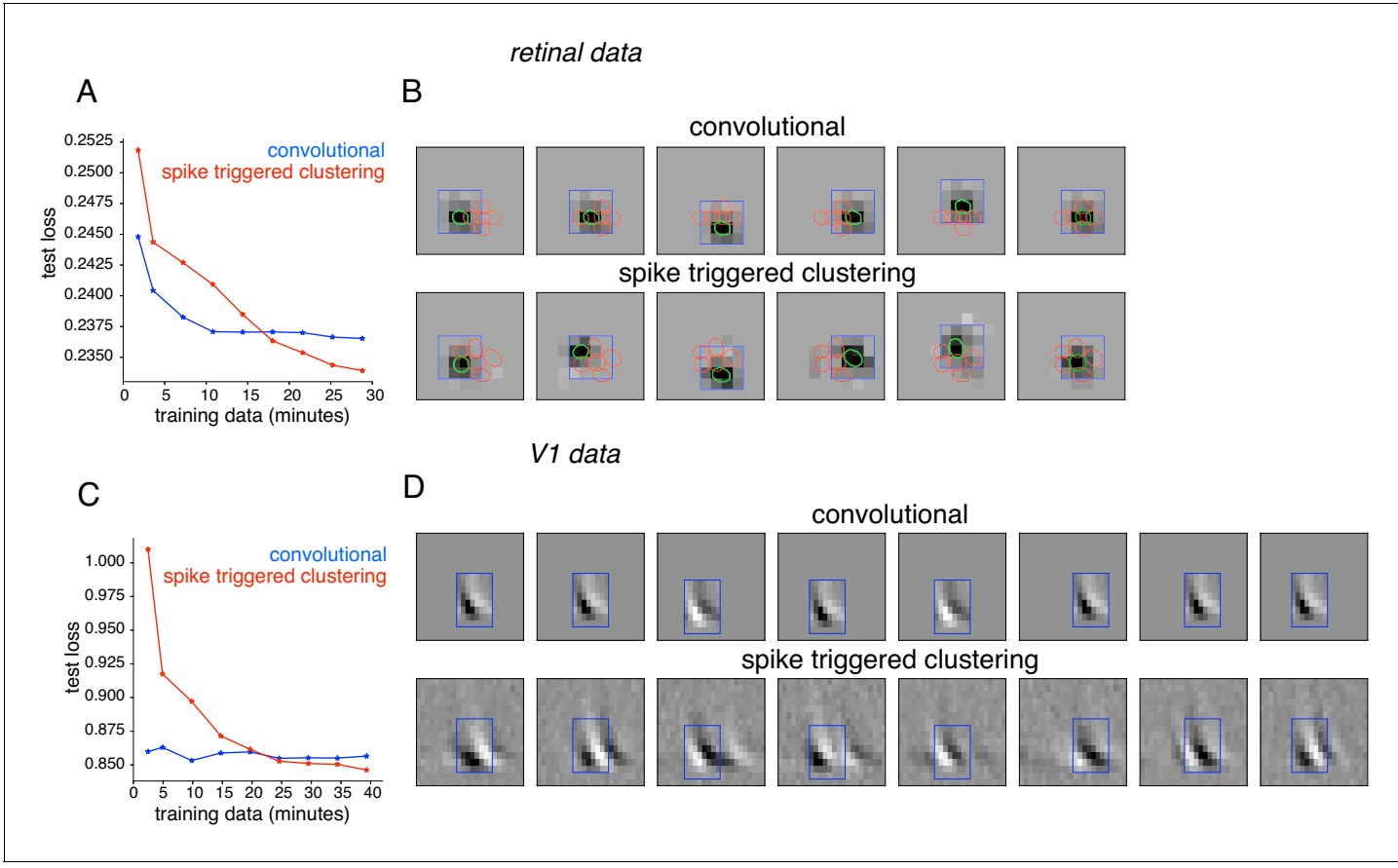

**Figure 10.** Comparison with convolutional subunit model. Subunits estimated for the retinal ganglion cell from *Figure 1A* using a convolutional model, similar to *Vintch et al. (2015)*. The visual stimulus is convolved with a 4 × 4 spatial filter, separately scaled for each location of filter, followed by exponentiation and weighted summation, resulting in Poisson firing rate. Parameters fit using gradient descent. (A) Comparison of neural response predictions for convolutional and spike-triggered clustering analysis. Negative log-likelihood loss on 3 min of test data (y-axis) is shown as a function of increasing duration of training data (x-axis), for the convolutional model (blue) and spike triggered clustering (with locally normalized L1 regularization, =0.1, no output nonlinearity, red). The convolutional model is more efficient (better performance with less training data), but spike triggered clustering performs better with more training data. (B) Estimated subunits for convolutional model (top row) and spike triggered clustering (bottom row, six subunits after cross-validation) for 24 min of training data. Six filters are presented for the convolutional model, translated at location of six strongest learned weights. Columns indicate overlapping subunits from the two methods, matched greedily based on the inner product of the spatio-temporal filters. (C) Similar to (A) for the complex cell from *Figure 8A*, evaluated on 5 min of test data (y-axis) with a convolutional model that has 8 × 8 spatio-temporal filter (same dimensions as *Vintch et al., 2015* and the spike triggered clustering model without regularization. (D) Matched subunits (similar to B) for convolutional model and spike triggered clustering (eight filters selected after cross-validation, see *Figure 8A*), estimated using 45 min of training data.

The online version of this article includes the following figure supplement(s) for figure 10:

**Figure supplement 1.** Comparison between spike triggered clustering and convolutional subunit model (*Vintch et al., 2015*) with quadratic nonlinearity.

The closest prior work in retina is spike-triggered non-negative matrix factorization (SNMF) (*Liu et al., 2017*; *Jia et al., 2018*). In addition to the greater efficiency of spike-triggered clustering (*Figure 9*), there are several technical differences worth noting. First, the assumptions of the SNMF algorithm are not directly matched to the structure of the subunit response model: the present model assumes nonnegative subunit outputs, while the SNMF factorization method assumes that the subunit filters are nonnegative. This assumption of nonnegative filters is inconsistent with suppressive surrounds previously reported in retinal bipolar cells (*Dacey et al., 2000*; *Fahey and Burkhardt, 2003*; *Turner et al., 2018*), and limits the application of the method to other systems such as V1 (*Figure 8*). In contrast, the spike-triggered clustering method makes no assumptions about the structure of the subunit filers. Second, the SNMF method requires a sequence of additional steps to

relate the estimated matrix factors to the parameters of an LNLN model. In contrast, in the present work the clustering of spike-triggered stimuli is shown to be equivalent to optimizing the likelihood of a LNLN model, eliminating extra steps in relating model parameters to biology. Third, the comparison to bipolar cell receptive fields using SNMF relied on a regularization parameter that trades off sparseness and reconstruction error. This regularization can influence the size of the estimated subunits, and thus the comparison to bipolar receptive fields. In contrast, the present method includes a novel regularizer that encourages spatial contiguity of receptive fields, while exerting minimal influence on the size of estimated subunits, and the strength of this regularizer is chosen using a cross-validation procedure.

Another study (*Freeman et al., 2015*) presented an approach to estimating subunits from OFF midget RGCs with visual inputs presented at single-cone resolution, and was used to obtain a successful cellular resolution dissection of subunits. However, this method assumed that subunits receive inputs from non-overlapping subsets of cones, which is likely to be incorrect for other cell types, including parasol RGCs. Moreover, the cone to subunit assignments were learned using a greedy procedure, which was effective for subunits composed of 1–2 cone inputs, but is more likely to converge to incorrect local minima in more general conditions.

Recent studies fitted a large class of LNLN models to simulated RGC responses (*McFarland et al., 2013*), to recorded ON-OFF mouse RGC responses (*Shi et al., 2019*), and to flickering bar responses in salamander RGCs (*Maheswaranathan et al., 2018*) using standard gradient descent procedures. These approaches have the advantage of flexibility and simplicity. In contrast, this work assumes a specific subunit nonlinearity that leads to a specialized efficient optimization algorithm, and applies it to recorded data from primate RGCs. The choice of an exponential subunit non-linearity enables likelihood maximization for Gaussian stimuli, which leads to more accurate estimates with limited data, and reduces the computational cost of each iteration during fitting because the loss depends only on the collection of stimuli that elicit a spike (*Ramirez and Paninski, 2014*). Additionally, the soft-clustering algorithm requires far fewer iterations than stochastic gradient descent algorithms used in fitting deep neural networks (*Kingma and Ba, 2014*; *Duchi et al., 2011*). Depending on the experimental conditions (e.g. type of stimuli, duration of recording, number of subunits to be estimated), the present approach may provide more accurate estimates of subunits, with less data, and more quickly.

Other recent studies applied convolutional and recurrent neural network models, fitted to natural scene responses, to mimic a striking variety of retinal response properties (*McIntosh et al., 2016*; *Batty et al., 2016*). The general structure of the model – convolutions and rectifying nonlinearities – resembles the subunit model used here. However, the number of layers and the interactions between channels do not have a direct correspondence to the known retinal architecture, and to date the specific properties of the inferred subunits, and their contributions to the behaviors of the model, have not been elucidated. The complexity of the underlying models suggest that a comparison to the biology may prove difficult.

In the context of V1 neurons, the present approach has advantages and disadvantages compared to previous methods. Spike-triggered covariance (STC) methods (*Rust et al., 2005*) estimate an orthogonal basis for the stimulus subspace captured by the subunits. However, the underlying subunits are unlikely to be orthogonal and are not guaranteed to align with the basis elements. The method also requires substantial data to obtain accurate estimates. To reduce the data requirements, other methods (*Eickenberg et al., 2012*; *Vintch et al., 2015*; *Wu et al., 2015*) imposed an assumption of convolutional structure on the subunits. This approach is efficient, because it effectively fits only one subunit filter that is then applied convolutionally. However, as shown above using recorded data from RGCs and V1 complex cells, the convolutional restriction may not be necessary when enough data are available, and under these conditions, the present method can more accurately reveal subunit structure in receptive fields that is not precisely convolutional (*Figure 10*).

A generalization of the STC approach (*Sharpee et al., 2004*; *Paninski, 2003*; *Eickenberg et al., 2012*) involves finding a subspace of stimulus space which is most informative about a neuron's response. This approach also yields a basis for the space spanned by subunits, but the particular axes of the space may or may not correspond to the subunits. Unlike the method presented here, this approach requires minimal assumptions about stimulus statistics, making it very flexible. However, methods based on information-theoretic measures typically require substantially more data than are available in experimental recordings (*Paninski, 2003*).

Although constraints on the structure of the subunit model (e.g. convolutional), and the estimation algorithm (e.g. spike-triggered covariance or clustering), both influence the efficiency of subunit estimation, the choice of priors used to estimate the subunits efficiently with limited data (i.e. regularization) is also important. In the present work, a regularizer was developed that imposes spatial contiguity, with minimal impact on the size of estimated subunits. A variety of other regularization schemes have been previously proposed for spatial contiguity for the estimation of V1 subunits (*Park and Pillow, 2011*). These could be incorporated in the estimation of the present model using a corresponding projection operator.

As suggested by the results, another algorithmic modification that could improve the subunit estimation is the hierarchical variant of the clustering approach (*Figure 2—figure supplement 1*). By splitting one of the subunits into two subunits at each step, models with different number of subunits can be estimated with greater efficiency, and preliminary exploration indicated that this may offer improvements in data efficiency and speed. However, potential tradeoffs emerging from the additional assumptions will require further study.

## Revealing the impact of non-linear computations using null stimuli

The null stimulus methodology revealed the failure of LN models, and isolated the nonlinear behaviors arising from receptive field subunits. Previous studies have developed specialized stimuli to examine different aspects of the cascaded linear-nonlinear model. Contrast reversing gratings (CRGs) have been the most commonly used stimuli for identifying the existence of a spatial nonlinearity (*Hochstein and Shapley, 1976*; *Demb et al., 2001*). Recent studies (*Schwartz et al., 2012*; *Turner and Rieke, 2016*) extended this logic to more naturalistic stimuli, with changes in responses to translations and rotations of texture stimuli revealing the degree of spatial nonlinearity. Another (*Bölinger and Gollisch, 2012*) developed a closed-loop experimental method to measure subunit nonlinearities, by tracing the intensities of oppositely signed stimuli on two halves of the receptive field to measure iso-response curves. More recent work (*Freeman et al., 2015*) developed stimuli in closed loop that targeted pairs of distinct cones in the receptive field, and used these to identify linear and nonlinear summation in RGCs depending on whether the cones provided input to the same or different subunits.

The null stimulus approach introduced here offers a generalization of the preceding methods. Null stimuli can be generated starting from any desired stimulus ensemble (e.g. natural images), and the method does not assume any particular receptive field structure (e.g. radial symmetry). Construction of the null stimuli does, however, require fitting a linear (or LN) model to the data in closed loop. Projecting a diverse set of stimuli (e.g. white noise) onto the null space yields a stimulus ensemble with rich spatio-temporal structure that generates diverse neural responses. By construction, the null stimulus is the closest stimulus to the original stimulus that is orthogonal to the receptive field, and thus minimally alters other statistical properties of the stimulus that affect neural response. This property is useful for studying the effect of spatial nonlinearities in the context of specific stimuli, such as natural scenes. Whereas contrast reversing gratings do not reveal stronger nonlinearities in OFF parasol cells compared to ON parasol cells in responses with natural stimuli (*Turner and Rieke, 2016*), the null stimulus captures these differences (*Figure 6*), highlighting the advantages of tailored stimulation with rich spatio-temporal responses.

## Further applications and extensions

The assumptions behind the proposed subunit model enable efficient fitting and interpretability, but also lead to certain limitations that could potentially be overcome. First, the choice of an exponential subunit nonlinearity improves efficiency by allowing the maximum expected likelihood approximation. However, other nonlinearities that allow this approximation, such as a rectified quadratic form (*Bölinger and Gollisch, 2012*), could be explored. Second, in fitting RGCs, the assumption of space-time separable subunits significantly reduces the number of parameters that need to be estimated, but could be relaxed with more data. The space-time separability assumption can be verified by checking if the spatio-temporal STA had rank one. In the present data (not shown), the STA has rank one when only pixels corresponding to the receptive field center were considered but had higher rank when the surround was also taken into consideration (*Enroth-Cugell and Pinto, 1970*; *Cowan et al., 2016*). Thus, replacing the space-time separability

assumption may be desirable. For example, a modification to the procedure that enforces subunits with rank two (not shown) could capture the center-surround structure of bipolar cell receptive fields (*Dacey et al., 2000*; *Turner et al., 2018*).

The methods developed here may also prove useful in tackling additional challenging problems in neural circuitry and nonlinear coding, in the retina and beyond. First, applying the model to high-resolution visual stimulation of the retina could be used to reveal how individual cones connect to bipolar cells, the likely cellular substrate of subunits, in multiple RGC types (see *Freeman et al., 2015*). Second, the incorporation of noise in the model, along with estimation of shared subunits (*Figure 4*), could help to probe the origin of noise correlations in the firing of nearby RGCs (see *Rieke and Chichilnisky, 2014*). Modeling the nonlinear interaction between subunits (*Kuo et al., 2016*; *Manookin et al., 2018*) could also improve prediction accuracy, especially for natural scenes. Third, extending the subunit model to include gain control, another ubiquitous nonlinear computation in the retina and throughout the visual system, could provide a more complete description of the responses in diverse stimulus conditions (*Heeger, 1992*; *Carandini and Heeger, 2012*). In addition to these potential scientific advances, a more accurate functional subunit model may be critical for the development of artificial retinas for treating blindness. Finally, the successful application of the method to V1 data (*Figure 8*) suggests that the method could be effective in capturing computations in other neural circuits that share the nonlinear subunit motif.

# Materials and methods

## Key resources table

| Reagent type (species) or resource | Designation | Source or reference | Identifiers | Additional information |
|---|---|---|---|---|
| Biological Sample | Macaque retina | UC Davis Primate Research Center | | |
| Biological Sample | Macaque retina | Stanford University | | |
| Biological Sample | Macaque retina | University of California Berkeley | | |
| Biological Sample | Macaque retina | Salk Institue | | |
| Biological Sample | Macaque retina | The Scripps Research Institute | | |
| Chemical compound, drug | Ames' medium | Sigma-Aldrich | Cat #1420 | |
| Software, algorithm | MGL | Gardner Lab | http://gru.stanford.edu/doku.php/mgl/overview | |
| Software, algorithm | MATLAB | Mathworks | | |
| Software, algorithm | Python | https://www.python.org/ | | |
| Software, algorithm | Intaglio | Purgatory Design | | |
| Software, algorithm | Custom spike sorting software | Chichilnisky Lab | | |

## Recordings

Detailed preparation and recording methods are described elsewhere (*Litke et al., 2004*; *Frechette et al., 2005*; *Chichilnisky and Kalmar, 2002*). Briefly, eyes were enucleated from seven terminally anesthetized macaque monkeys (Macaca sp.) used by other experimenters in accordance with institutional guidelines for the care and use of animals. Immediately after enucleation, the

anterior portion of the eye and the vitreous were removed in room light. The eye was stored in darkness in oxygenated Ames' solution (Sigma, St Louis, MO) at 33°C, pH 7.4. Segments of isolated or RPE-attached peripheral retina (6–15 mm temporal equivalent eccentricity [*Chichilnisky and Kalmar, 2002*], approximately 3 mm x 3 mm) were placed flat, RGC side down, on a planar array of extracellular microelectrodes. The array consisted of 512 electrodes in an isosceles triangular lattice, with 60 μm inter-electrode spacing in each row, covering a rectangular region measuring 1800 μm x 900 μm. While recording, the retina was perfused with Ames' solution (35°C for isolated recordings and 33°C for RPE-attached recordings), bubbled with 95% $O_2$% and 5% $CO_2$, pH 7.4. Voltage signals on each electrode were bandpass filtered, amplified, and digitized at 20 kHz.

A custom spike-sorting algorithm was used to identify spikes from different cells (*Litke et al., 2004*). Briefly, candidate spike events were detected using a threshold on each electrode, and voltage waveforms on the electrode and nearby electrodes in the 5 ms period surrounding the time of the spike were extracted. Candidate neurons were identified by clustering the waveforms using a Gaussian mixture model. Candidate neurons were retained only if the assigned spikes exhibited a 1 ms refractory period and totaled more than 100 in 30 min of recording. Duplicate spike trains were identified by temporal cross-correlation and removed. Manual analysis was used to further select cells with a stable firing rate over the course of the experiment, and with a spatially localized receptive field.

## Visual stimuli

Visual stimuli were delivered using the optically reduced image of a CRT monitor refreshing at 120 Hz and focused on the photoreceptor outer segments. The optical path passed through the mostly transparent electrode array and the retina. The relative emission spectrum of each display primary was measured with a spectroradiometer (PR-701, PhotoResearch) after passing through the optical elements between the display and the retina. The total power of each display primary was measured with a calibrated photodiode (UDT Instruments). The mean photoisomerization rates for the L, M, and S cones were estimated by computing the inner product of the display primary power spectra with the spectral sensitivity of each cone type, and multiplying by the effective collecting area of primate cones (0.6 μm$^2$) (*Angueyra and Rieke, 2013*; *Schnapf et al., 1990*). During white noise and null stimulus, the mean background illumination level resulted in photoisomerization rates of (800–2200, 800–2200, 400–900) for the (L, M, S) cones. The pixel size was either 41.6 microns (eight monitor pixels on a side) or 20.8 microns (four monitor pixels on a side). New black binary white noise frames were drawn at a rate of 60 Hz (*Figure 2*) or 30 Hz (*Figures 5,6*). The pixel contrast (difference between the maximum and minimum intensities divided by the sum) was either 48% or 96% for each display primary, with mean luminance at 50%.

The details of natural stimuli used are presented in *Heitman et al. (2016)*. Briefly, the stimuli consisted of images from the Van Hateren database, shown for one second each, jittered according to eye movement trajectories recorded during fixation by awake macaque monkeys (Z.M. Hafed and R. J. Krauzlis, personal communication), (*Heitman et al., 2016*; *van Hateren and van der Schaaf, 1998*). Image intensities produced mean photoisomerization rates of (1900, 1800, 700) for the (L, M, S) cones, respectively. The pixel width was 10.4 microns (two monitor pixels on a side), and image frames refreshed at 120 Hz. The training data, consisting of 59 groups of 60 distinct natural scenes, was interleaved with identical repeats of testing data, consisting of 30 distinct natural scenes.

## Subunit estimation

The light responses of RGCs were modeled using a cascade of two linear-nonlinear (LN) stages, followed by a Poisson spike generator. Specifically, the instantaneous stimulus-conditional firing rate was modeled as $\lambda_t | X_t = g(\sum_n w_n \exp(K_n^T X_t))$ where $X_t$ is the visual stimulus at time $t$ (expressed as a vector), $K_n$ are the spatial subunit filters (also vectors), $w_n$ are non-negative weights on different subunits, and $g(x) = \frac{x^a}{bx+1}$, $(b \geq 0)$ is the output nonlinearity.

The parameters $\{K_n, w_n, a, b\}$ are estimated by minimizing their negative log-likelihood given observed spiking responses $Y_t$ to visual stimuli $X_t$, in two steps. In the first step, the parameters $\{K_n, w_n\}$ are estimated using a clustering procedure while ignoring the nonlinearity ($g$). In the second step, $\{g, w_n\}$ are estimated using gradient descent, with the $K_n$ held fixed (up to a scale factor).

The clustering procedure is derived by minimizing an approximate convex upper bound of the negative log-likelihood based on current parameter estimates $\{K_n^0, b_n^0\}$. The convex upper-bound is derived as follows:

$$\mathcal{L}(w_n, K_n) = \frac{1}{T}\sum_{t=1}^{t=T}\lambda_t - \frac{1}{T}\sum_{t=1}^{t=T}Y_t\log(\lambda_t) \tag{1}$$

$$\approx \mathbb{E}(\lambda) - \frac{1}{T}\sum_{t=1}^{t=T}Y_t\log(\lambda_t) \tag{2}$$

$$= \sum_{n=1}^{n=N}w_n e^{K_n^T K_n/2} - \frac{1}{T}\sum_{t=1}^{t=T}Y_t\log(\sum_{n=1}^{n=N}w_i e^{K_n^T X_t}) \tag{3}$$

$$\leq \sum_{n=1}^{n=N}w_n e^{K_n^T K_n/2} - \frac{1}{T}\sum_{t}\sum_{n=1}^{n=N}Y_t\,\alpha_{t,n}^0[(K_n - K_n^0)^T X_t(\log(w_n) - \log(w_n^0))] + c(K_n^0, w_n^0)] \tag{4}$$

$$= \sum_{n=1}^{n=N}(w_n e^{K_n^T K_n/2} - \frac{1}{T}\sum_{t}Y_t\,\alpha_{t,n}^0[(K_n - K_n^0)^T X_t(\log(w_n) - \log(w_n^0))] + c(K_n^0, w_n^0)) \tag{5}$$

where $\alpha_{t,n}^0 = \frac{w_n^0 e^{K_n^0 \cdot X_t}}{\sum_{m=1}^{m=N}w_j^0 e^{K_m^0 \cdot X_t}}$ and $c(K_n^0, w_n^0)$ does not depend on parameters $\{K_n, w_n\}$. For simplicity, we present the equations for a standard Gaussian stimulus ($X_t \sim \mathcal{N}(0, I)$) (modifications for arbitrary Gaussian stimuli are straightforward). Since the first term in the log-likelihood (*Equation 1*) does not depend on recorded responses ($Y_t$), it is replaced with its expected value across stimuli (*Equation 2*). Because the projections of the input distribution onto filters $K_n$ are assumed to be Gaussian, this expectation can be computed using the moment generating function of a Gaussian distribution (*Equation 3*). The second term only depends on the stimuli for which the cell generated spikes ($Y_t > 0$), reducing computational cost. Finally, to optimize the resulting non-convex function, a convex upper bound was minimized at each step. Specifically, the second term is replaced by a first-order Taylor approximation (*Equation 4*) (the first term is already convex). This upper bound is separable across parameters of different subunits, as can be seen by re-arranging the summation over time and subunits (*Equation 5*). Therefore, the parameters for each subunit can be updated separately by minimizing the corresponding term in *Equation 5*.

The parameters for each subunit are optimized in three steps. First, the subunit activations, $\alpha_{t,n}$ are updated based on the parameters $\{K_n, w_n\}$ from the previous iteration. Then, setting to zero the partial derivatives of *Equation 5* with respect to $K_n$ and $w_n$ produces a system of two equations, which are rearranged to yield the remaining two steps:

1. Update subunit activations (cluster weights) for each stimulus : $\alpha_{t,n} \leftarrow \frac{w_n e^{K_n \cdot X_t}}{\sum_{m=1}^{m=N}w_m e^{K_m \cdot X_t}}$.

2. Update linear subunit filters (cluster centers) : $K_n \leftarrow \frac{\sum_t Y_t \alpha_{t,n} X_t}{\sum_t Y_t \alpha_{t,n}}$.

3. Update relative weights for different subunits : $w_n \leftarrow \frac{\sum_t Y_t \alpha_{t,n}}{T}e^{-\frac{1}{2}K_n^T K_n}$.

Hence, applying the three steps repeatedly results in a clustering algorithm that iteratively minimizes a convex upper bound on the approximate negative log-likelihood function. Since the upper bound is tight at the current parameter estimates, the successive minimization of the convex upper bound reduces the approximate negative log-likelihood monotonically, leading to guaranteed convergence. However, due to non-convexity of the approximate negative log-likelihood function, the estimated parameters might not correspond to a global minimum. In the limit of large recording duration, the approximation in *Equation 2* is exact.

The memory requirement for each step of the algorithm is proportional to the number of spike triggered stimuli, which could be large for fine resolution stimuli and/or high firing rates. Computational runtime depends linearly on the number of spikes, the stimulus dimensions, the number of

subunits, and the number of steps for convergence of the clustering procedure. For most of the results presented in this paper, each fit converged in ∼ 100 steps and required a few minutes on a single-core computer. Additional speed improvements would be possible by parallelizing each step of clustering across subunits, making runtime independent of the number of subunits. The resulting subunits can be interpreted as a decomposition of the spike triggered average (STA), since the sum of estimated subunits ($K_n$), weighted by their expected contribution to the response ($w_n e^{\frac{1}{2}K_n^T K_n}$), is proportional to the spike-triggered average:

$$\sum_{n \in [N]} (w_n e^{\frac{1}{2}K_n^T K_n}) K_n = \sum_{n \in [N]} \left( \frac{\sum_t Y_t \alpha_{t,n}}{T} \right) K_n \tag{6}$$

$$= \sum_t \sum_{n \in [N]} \frac{Y_t \alpha_{t,n} X_t}{T} \tag{7}$$

$$= \frac{\sum_t Y_t X_t}{T} \tag{8}$$

where (6) comes form (3) (at convergence), (7) comes from (2) (at convergence), and (8) arises because the $\alpha_{t,n}$ sum to one. Step (6) may be interpreted as replacing the average spike rate, multiplied by the contribution of each subunit. Step (7) may be interpreted as a weighted average of spike-triggered stimuli, where the weighting assigns responsibility for each spike across the subunits. The above formulae ignore the output non-linearity, justified by the observation that it is approximately linear after fitting.

## Incorporating priors

The log-likelihood can be augmented with a regularization term, to incorporate prior knowledge about the structure of subunits, thereby improving estimation in the case of limited data. Regularization is incorporated into the algorithm by projecting the subunit kernels ($K_n$) with a proximal operator ($P_\lambda$), derived from the regularization function, after each update of the cluster centers (*Figure 11*). For a regularization term of the form $\lambda f(K)$, the proximal operator is defined as $\mathcal{P}_{\lambda,f}(K^0) = \text{argmin}_K (f(K) + (1/2\lambda) \|K - K^0\|_2^2)$. Regularization was applied for estimating subunits for both single cell (*Figure 3*) as well as population (*Figure 4*) data and in each case, the regularization strength ($\lambda$) was chosen by cross-validation (see below).

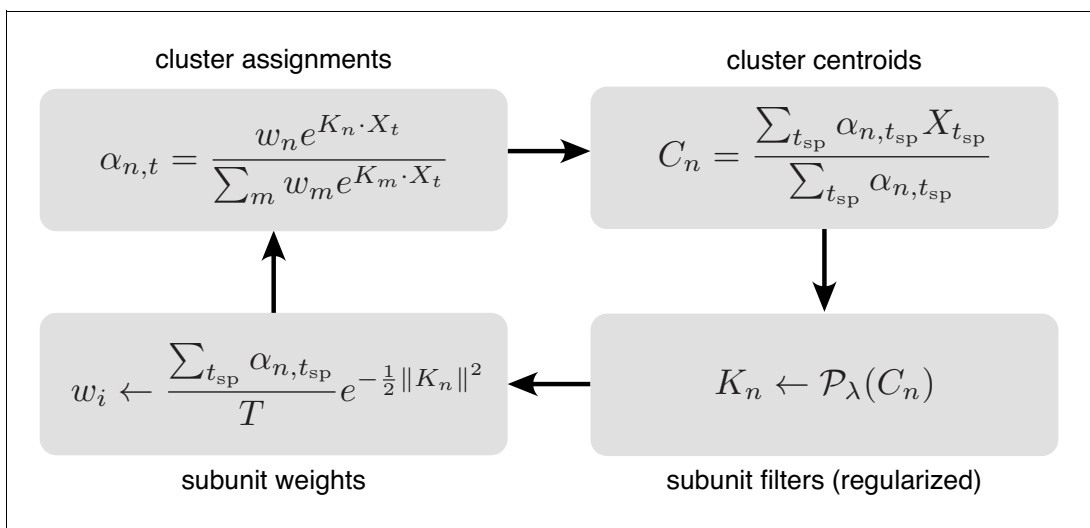

**Figure 11.** Iterative fitting of subunits, partitioned into four steps. The subunit kernels ($K_n$) and weights ($w_n$) are randomly initialized, and used to compute soft cluster assignments ($\alpha_{n,t}$ upper left), followed by cluster centroid computation ($C_n$ - upper right), estimation of subunit kernels ($K_n$ - lower right) and subunit weights ($w_n$ - lower left). The summations are only over times when the cell generated a spike ($t_{sp}$).

For $L_1$-norm regularization (section 3), $f(K) = \sum_i |K^i|$, where $K^i$ is the value of $i$th pixel, and the proximal operator is a soft-thresholding operator $\mathcal{P}_{\lambda,|.|}(K) = \max(K - \lambda, 0) - \max(-K - \lambda, 0)$. For the locally normalized $L_1$-norm regularization, the proximal operator of the Taylor approximation was used at each step. The Taylor approximation is a weighted $L_1$-norm with the weight for each pixel determined by the parameter value of the neighboring pixels. Hence, the proximal operator for the $i$th pixel is $\mathcal{P}_{\lambda,|.|}(K^i) = \max(K^i - a^i\lambda, 0) - \max(-K^i - a^i\lambda, 0)$, where $a^i = 1/(\epsilon + \sum_{j \in \text{Neighbors}(i)} |K^j|)$ for $\epsilon = 0.01$ black (smaller than typical non-zero values in $K$, which are on the order of unity). Simulations were used to verify that this regularizer induces a preference for spatially contiguous subunits, while being relatively insensitive to subunit area (compared with $L_1$ regularization).

The entire procedure is summarized in *Figure 11*, with simplified notation illustrating the four key steps.

## Subunit estimation using hierarchical clustering

As shown in *Figure 2*, a hierarchical organization of subunits was observed in fits to RGC data, with one subunit broken down into two subunits each time the total number of subunits was increased by one. This hierarchical structure can be enforced explicitly to make the estimation procedure more efficient.

Since the softmax weights $\alpha_m$ (between 0 and 1) can be interpreted as the 'activation probability' of subunit $m$, hierarchical clustering is performed by estimating two 'child' subunits $m_1, m_2$ with factorized activation probability : $\alpha_{m_1} = \alpha_m \alpha_{m_1|m}$, where $\alpha_{m_1|m}$ is the conditional probability of child $m_1$ given the activation of parent subunit with $\alpha_{m_1|m} = \frac{e^{K_{m_1}^T X + b_{m_1}}}{e^{K_{m_1}^T X + b_{m_1}} + e^{K_{m_2}^T + b_{m_2}}}$. The factorization is accurate if the activation of the parent subunit is the sum of activation of the child subunits ($e^{K_m.X + b_m} = e^{K_{m_1}.X + b_{m_1}} + e^{K_{m_2}.X + b_{m_2}}$).

The child subunits were initialized by adding independent random noise to parent subunits ($K_{m_i} = K_m + \epsilon_i$) and equally dividing the subunit weight into two ($e^{b_{m_i}} = \frac{e^{b_m}}{2}$). Hierarchical clustering was performed by iteratively updating the cluster assignments ($\alpha_{m_i}$) and cluster centers ($K_{m_i}$, $b_{m_i}$) for the child subunits using the factored $\alpha$ as outlined above.

The parent subunit for splitting was chosen greedily, to yield maximum increase in log-likelihood at each step. Hierarchical subunit estimation provides a computationally efficient way to estimate the entire solution path with different $N$. The estimated subunits obtained with this procedure are shown in *Figure 2—figure supplement 1*.

## Population model

In Section 4, a method to estimate a common bank of subunits jointly using multiple nearby cells is described. For each cell $c$, the firing rate is given by $\lambda_{c,t} = g_c \left( \sum_{n=1}^{n=N} w_{n,c} e^{K_n.X_t} \right)$ where $w_{n,c}$ are the cell-specific subunit weights. The model parameters were estimated by maximizing the summation of log-likelihood across cells.

Similar to the single-cell case, the estimation was performed in two steps: 1) Ignoring the output nonlinearity $g_c$, estimate $\{K_n, w_{n,c}\}$ by clustering, and 2 ) Estimate $g_c$, $w_{n,c}$ and the magnitude of $K_n$ for each cell independently using gradient descent, with the vector direction of $\{K_n\}$ fixed. Clustering in the first step can be interpreted as finding a common set of centroids to cluster the spike-triggered stimuli for multiple cells simultaneously. The following steps are repeated iteratively:

1. Update subunit activations (cluster weights) for each stimulus and cell: $\alpha_{t,n,c} \leftarrow \frac{w_{n,c} e^{K_n.X_t}}{\sum_{m=1}^{m=N} w_{j,c} e^{K_m.X_t}}$

2. Update linear subunit filters (cluster centers) using population responses: $K_n \leftarrow \mathcal{P}_\lambda \left( \frac{\sum_t \sum_c Y_{t,c} \alpha_{t,n,c} X_t}{\sum_t \sum_c Y_{t,c} \alpha_{t,n,c}} \right)$

3. Update subunit weights ($w_{n,c}$) for each cell: $w_{n,c} \leftarrow \frac{\sum_t \alpha_{t,n,c}}{T} e^{-\frac{1}{2} K_n^T K_n}$

As with subunit estimation using single-cell responses, prior knowledge about subunit structure can be incorporated with a proximal operator ($\mathcal{P}(.)$) in the clustering loop. For RGC data, locally normalized L1 regularization was used (*Figure 4*).

## Application to neural responses

The subunit estimation algorithm was applied to a rectangular segment of stimulus covering the receptive field (along with a one stimulus pixel boundary around it) and to the RGC spike counts over time (8.33 ms bin size). The binary white noise stimulus was temporally filtered and the resulting normally distributed stimulus (zero mean) was used for subunit estimation ($X_t$). The receptive field was estimated as a contiguous set of pixels black in the STA with magnitude larger than 2.5$\sigma$, where $\sigma$ is the standard deviation of the background noise in the STA. The time course for filtering (125 ms length) was estimated by averaging the time course of significant pixels in the STA. The data were partitioned into three groups: testing data consisted of a contiguous segment of recording (last 10%), and the rest of the recording was randomly partitioned for training (90%) and validation (10%) data.

Models were fit with different numbers of subunits $N$ (1-12) and regularization values $\lambda$. Selection of hyperparameters ($N$ and $\lambda$) was performed by averaging the performance of multiple fits with different training/validation partitions and random initializations (see Figure captions for details). For a given number of subunits, the regularization value was chosen by optimizing the performance on validation data. Similarly, the most accurate model was chosen by optimizing over the number of subunits as well as regularization strength. When data with repeated presentations of the same stimulus were available (e.g. *Figures 6,7* ), the prediction accuracy was evaluated by measuring the correlation between peri-stimulus time histogram (PSTH) of recorded data and the predicted spikes for the same number of repetitions. The PSTH was computed by averaging the discretized responses over repeats and gaussian smoothing with standard deviation 16.67 ms.

For null stimuli, responses from 30 repeats of a 10 s long stimulus were divided into training (first 5 s) and testing (last 5 s) data. Since the training data were limited, the subunit filters and weights (phase 1) were first estimated from responses to a long white noise stimulus and only the subunit scales and output nonlinearity (phase 2) were estimated using the null stimulus.

For natural scenes stimuli, the spatial kernel for subunits was first estimated from responses to the white noise stimulus (phase 1), and the scales of subunits and output nonlinearity were then fit using responses to a natural stimulus (phase 2). This procedure was used because subunits estimated directly from natural scenes did not show any improvement in prediction accuracy, and all the subunits were essentially identical to the RF, presumably because of the strong spatial correlations in natural images.

V1 neurons were fit to previously published responses to flickering bars (see *Rust et al., 2005* for details). In contrast to the retinal data, the two dimensions of the stimulus were time and one dimension of space (orthogonal to the preferred orientation of the cell). The estimation procedure (without regularization) was applied for different numbers of subunits. For successive numbers of subunits ($N$ and $N + 1$), the hierarchical relationship was evaluated by greedily matching $N - 1$ filters across the two fits using the inner product of their spatio-temporal filters and splitting the remaining subunit in the $N$-subunit fit into the two remaining subunits in the $N + 1$-subunit fit. The maximum number of subunits was chosen by cross validation as described above.

## Simulated RGC model

To test the estimation procedure in conditions where the answer is known, the procedure was applied to responses generated from a simulated RGC (Appendix I). In the simulation, each RGC received exponentiated inputs from bipolar cells, which in turn linearly summed inputs from photoreceptors. The spatial dimensions of the simulation and the number of cones and bipolar cells were approximately matched to parasol RF centers at the eccentricities of the recorded data (*Schwartz and Rieke, 2011*; *Jacoby et al., 2000*). Below, all the measurements are presented in grid units (g.u.), with 1 g.u. = 1 micron resulting in a biologically plausible simulation. The first layer consisted of 64 photoreceptors arranged on a jittered hexagonal lattice with nearest-neighbor distance 5 g.u., oriented at 60˚ with respect to the stimulus grid. The location of each cone was independently jittered with a radially symmetric Gaussian with standard deviation black 0.35 g.u. Stimuli were pooled by individual photoreceptors with a spatio-temporally separable filter, with time course derived from a typical parasol cell and a Gaussian spatial filter (standard deviation = black 1 g.u.).

The second layer of the cascade consisted of 12 model bipolar cells, each summing the inputs from a spatially localized set of photoreceptors, followed by an exponential nonlinearity. The

photoreceptors were assigned to individual bipolars by k-means clustering (12 clusters) of their locations. Finally, the RGC firing rate is computed by a positively weighted summation of bipolar activations. The bipolar weights were drawn from a Gaussian distribution with standard deviation equal to 10% of the mean. The choice of photoreceptors and bipolar cell weights give a mean firing rate of ≈ 19 spikes/s.

First, the subunit model was fitted to 24 min of responses to a white noise stimulus, such that the receptive field was roughly covered by 6 × 6 stimulus pixels (Figure S1B), as in the recorded experimental data (*Figure 2B*). Subsequently, the subunit model was fitted to 6 hr of stimuli that excites each cone individually (Figure S1C). In this case, subunits are represented as weights on individual photoreceptors. In a retinal preparation, individual cone activation is possible using a fine resolution white noise such that the locations of individual photoreceptors can be identified (see *Freeman et al., 2015*).

## Null stimulus

Null stimuli were constructed to experimentally verify the nonlinear influence of estimated subunits on light response. Below, a general approach is first presented to generate a spatio-temporal null stimulus which can be used for any neural system. Subsequently, it is shown that if the STA is space-time separable, the spatio-temporal null stimulus can be derived with a simpler and faster method that only uses spatial information . Since this separability assumption is approximately accurate for RGCs, the spatial null stimulus is used for the results presented in main paper (*Figures 5,6*).

The purpose of spatio-temporal nulling is to find the closest stimulus sequence such that convolution with a specific spatio-temporal linear filter gives 0 for all time points. For simplicity, the algorithm is described only for a single cell (the extension to multiple cells is straightforward). Let $a(x,y,t)$ represent the spatio-temporal linear filter, estimated by computing the STA on white noise responses. The orthogonality constraint for a null stimulus $S(x,y,t)$ is written as:

$$\sum_{x,y}\sum_{\tau} a(x,y,\tau)S(x,y,t-\tau)=0, \ \ \forall t \in \{0\cdots T\}$$

The constraints for successive frames are not independent, but they become so when transformed to the temporal frequency domain. Writing $a(.,.,\omega)$ and $S(.,.,\omega)$ for the Fourier transform of $a(.,.,t)$ and $S(.,.,t)$ respectively, the constraints may be rewritten as:

$$\sum_{x,y} a(x,y,\omega)S(x,y,\omega)=0, \ \ \forall \omega \in \{\frac{2\pi i}{T}; i \in \mathbb{Z}\}$$

Hence for each temporal frequency, the spatial content can be projected onto the subspace orthogonal to the estimated spatial receptive field. Specifically, for a given temporal frequency $\omega$, given a vectorized stimulus frame $S_w$ and a matrix $A_\omega$ with columns corresponding to the vectorized receptive field, the null frame is obtained by solving the following optimization problem:

$$S_n = \mathrm{argmin}_S \frac{1}{2}\|S - S_w\|_2^2, \ \ \text{such that } A_\omega^T S = 0.$$

The solution may be written in closed form:

$$S_n = (I - A_\omega(A_\omega^T A_\omega)^{-1} A_\omega^T)S_w.$$

For the results presented in this paper, binary white noise stimuli ($S_w$) are projected onto the null space. However, in general, any visual stimulus can be used instead of white noise. The receptive field (linear filter) for each cell was estimated by computing a space-time separable approximation of the spike-triggered average (STA), with the support limited to the pixels with value significantly different from background noise (absolute value of pixel value > 2.5 σ, with σ a robust estimate of standard deviation of background noise).

In addition to the orthogonality constraint, two additional constraints are imposed: a) each pixel must be in the range of monitor intensities (−0.5 to 0.5) and b) the variance of each pixel over time must be preserved, to avoid the possibility of contrast adaption in photoreceptors (*Clark et al., 2013*, but see *Rieke, 2001*). These two constraints were incorporated using Dykstra's algorithm

(*Boyle and Dykstra, 1986*), which iteratively projects the estimated null stimulus into the subspace corresponding to the individual constraints until convergence. Setting the contrast of initial white noise black ($S_w$) to a value lower than 100% was necessary to satisfy these additional constraints. Finally, the pixel values of the resulting null stimuli were discretized to 8-bit integers, for display on a monitor with limited dynamic range.

For a space-time separable (rank one) STA (as is approximately true for RGCs), spatio-temporal nulling gives the same solution as spatial nulling. To see this, assume $a(x, y, t) = a_1(x, y)a_2(t)$, a space-time separable linear filter. Hence, the null constraint can be rewritten as

$$\sum_{x,y} \sum_{\tau} a_1(x, y)a_2(\tau)S(x, y, t - \tau) = \sum_{\tau} a_2(\tau)l(t - \tau) = 0 \ \forall t$$

where $l(t - \tau) = \sum_{x,y} a_1(x, y)S(x, y, t - \tau)$. In the frequency domain this is written as

$$a_2(\omega)l(\omega) = 0 \ \forall \omega$$

Since $a_2(t)$ has limited support in time, it has infinite support in frequency. Hence, $l(\omega) = 0 \ \forall \omega$ implying spatial-only nulling $l(t) = 0 \ \forall t$.

## Additional information

### Funding

| Funder | Grant reference number | Author |
|---|---|---|
| National Science Foundation | NSF IGERT Grant 0801700 | Nora Jane Brackbill |
| National Eye Institute | NIH NEI R01-EY021271 | EJ Chichilnisky |
| Howard Hughes Medical Institute | | Eero Simoncelli |
| Pew Charitable Trusts | | Alexander Sher |
| National Eye Institute | NIH NEI F31EY027166 | Colleen Rhoades |
| National Science Foundation | NSF GRFP DGE-114747 | Colleen Rhoades |
| National Eye Institute | NIH NEI P30-EY019005 | E J Chichilnisky |

The funders had no role in study design, data collection and interpretation, or the decision to submit the work for publication.

### Author contributions

Nishal P Shah, Conceptualization, Resources, Data curation, Software, Formal Analysis, Validation, Investigation, Visualization, Methodology, Writing - original draft, Writing - review and editing; Nora Brackbill, Colleen Rhoades, Alexandra Kling, Georges Goetz, EJ Chichilnisky, Experimentation; Alan M Litke, Resources, Software; Alexander Sher, Eero P Simoncelli, Conceptualization, Resources, Formal Analysis, Validation, Investigation, Methodology, Writing - original draft, Writing - review and editing, Supervision

### Author ORCIDs

Nishal P Shah (iD) https://orcid.org/0000-0002-1275-0381
Nora Brackbill (iD) https://orcid.org/0000-0002-0308-1382
Alan M Litke (iD) https://orcid.org/0000-0003-3973-3642
Eero P Simoncelli (iD) https://orcid.org/0000-0002-1206-527X
EJ Chichilnisky (iD) https://orcid.org/0000-0002-5613-0248

### Ethics

Animal experimentation: Eyes were removed from terminally anesthetized macaque monkeys (Macaca mulatta, Macaca fascicularis) used by other laboratories in the course of their experiments, in accordance with the Institutional Animal Care and Use Committee guidelines. All of the animals

were handled according to approved institutional animal care and use committee (IACUC) protocols (#28860) of the Stanford University. The protocol was approved by the Administrative Panel on Laboratory Animal Care of the Stanford University (Assurance Number: A3213-01).

## Decision letter and Author response

Decision letter https://doi.org/10.7554/eLife.45743.sa1
Author response https://doi.org/10.7554/eLife.45743.sa2

## Additional files

### Supplementary files

• Transparent reporting form

### Data availability

Data has been deposited with Dryad at: https://doi.org/10.5061/dryad.dncjsxkvk. Code available on Github: https://github.com/Chichilnisky-Lab/shah-elife-2020 (copy archived at https://github.com/elifesciences-publications/shah-elife-2020).

The following dataset was generated:

| Author(s) | Year | Dataset title | Dataset URL | Database and Identifier |
|---|---|---|---|---|
| Shah NP, Brackbill NJ, Rhoades C, Kling A, Goetz G, Litke AM, Sher A, Simoncelli E, Chichilnisky EJ | 2020 | Data from: Inference of nonlinear receptive field subunits with spike-triggered clustering | https://doi.org/10.5061/dryad.dncjsxkvk | Dryad Digital Repository, 10.5061/dryad.dncjsxkvk |

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
