## [Decision Letter]

Thank you for submitting your article "Inference of nonlinear spatial subunits in primate retina with Spike-Triggered Clustering" for consideration by *eLife*. Your article has been reviewed by Joshua Gold as the Senior Editor, a Reviewing Editor, and three reviewers. The reviewers have opted to remain anonymous.

The reviewers have discussed the reviews with one another and the Reviewing Editor has drafted this decision to help you prepare a revised submission.

Summary:

This manuscript introduces a new method for the estimation of nonlinear spatial subunits in neural data using an LNLN model (a cascade of two LN stages). The paper has clear merits by showing that it is beneficial to utilize subunit models for predicting parasol RGC responses and V1 neural responses to naturalistic stimuli. Parasol spike data has been previously mostly analyzed without taking this into account.

The main concern to address is that for a "tools and resources" papers needs to include clear benchmarking of the advantages of the current method against other existing and/or published approaches for subunit identification. The consensus among reviewers was that one would need to include comparison with at least one other method for both the retina and the V1 dataset. Some of the relevant methods that you could use are provided below, which offer different trade-offs between accuracy and ease of implementation and the amount of background work needed to set the hyperparameters. It would be helpful if the authors could include a discussion to justify their choice of the alternative method. The alternative method might be different for the V1 and the retina dataset. Including a more detailed discussion of the available methods and the authors' perspective on which would work best at each circumstance or generalize across the light/stimulus conditions would be very helpful to the field.

The second concern was about the apparent inconsistencies in how the steps of the proposed algorithm were chosen (as detailed below) and that more methodological details need to be provided.

Comparison with existing methods:

While the manuscript includes some discussion of the existing methods, it is still not clear which model is the best taken into account that the subunits of the current model do not correlate with the actual bipolar cell receptive fields but rather with their clusters in some way. For the reader to take up this method, one would need to clearly see the pros and cons in speed, resolution, easiness of implementation, neural correlates of subunits and/or other key features of this model against other nonlinear subunit models (e.g. Liu et al., 2017 and the simpler earlier version by Kaardal et al., 2013) and/or simple convolution neural network models (Eickenberg et al., 2013, Vintch et al., 2015). It is critical for the reader to be able to assess the value of the current model. Assuming this is done properly, and these clarifications are added, this paper would be significantly improved.

One would like to understand and document clearly performance of the subunit-identification method across stimulus conditions (white noise vs. naturalistic stimuli) against conventional currently used methods. This could perhaps be done by applying the new model to existing datasets of parasol RGCs in different experimental conditions. If such data are not available, then perhaps the method could be tested using simulated responses constructed to mimic properties of the retinal neurons.

Questions about the proposed methodology:

There are some inconsistencies in the design of the proposed methodology. For examples, the method is motivated with an example that involves hierarchical splitting of subunits, but this hierarchical version is actually not used further on. Then the bare method (without regularization and fairly coarse stimuli) is used to analyze populations of OFF parasol cells (Figure 2B-F), but Figure 3 shows that much better subunits are obtained with regularization and finer white noise. One wonders why this improved method wasn't used to obtain, for example, the distribution of subunit numbers. Further analyses are then again (partly) performed without regularization (e.g. joint fitting of multiple cells, V1 data). In particular when discussing the V1 data, where the subunits have very different structure than in the retina, one wonders whether regularization still works in the same way. Also, it is emphasized how the optimal number of subunits can be determined, but some of the subsequent analyses are performed for a fixed number of subunits (e.g. Figure 2 and Figure 3).

There are also several other parts where that information is difficult to extract or missing, which hinders understanding of the material.

The description of the subunit estimation approach via approximate maximum likelihood (subsection “Subunit estimation”) is central to the manuscript but quite difficult to follow. Some terms in the equation are not defined (K and w with superscript 0, c(K,w)), and the explanations for the derivations are cursory or absent. For example, when Equations 1-5 are derived, there is no mentioning of an iterative procedure (this only comes later; at this point, this seems to be only a reformulation of the log-likelihood), but some manipulations and terms here only make sense in the context of an iterative estimation procedure. This needs to be disentangled.

Relatedly, it seems important to spell out how the clustering algorithm (Equations 6-8) solves the maximum likelihood problem, in particular since this connection is central to the manuscript.

The application of the method to V1 data makes an important point regarding the general applicability of the method. This is nice, but only very little analysis is presented. Maybe the authors could at least show how the model performs for different numbers of subunits as in other analyses, to see how well the simple cell and the complex cell are separated in this respect. Also, the effect of regularization would be interesting here.

This assumption of space-time separability needs further discussion and justification. This assumption might be necessary to reduce the number of parameters to manageable levels, but it would be helpful if the authors could offer guidelines for when it might be appropriate.

About the response prediction to naturalistic stimuli. The use of white noise stimuli rather than natural stimuli might provide a good approximation for estimating of subunit filters, However, bipolar cells, which provide inputs to parasol RGCs, are strongly affected by stimulus correlation through gap-junction connectivity generating a nonlinear enhancement of bipolar activity, which could explain the differences found with real activity (Kuo et al., 2016, Manookin et al., 2018). These elements should be included in the Discussion section as one of the potential pitfalls of the approach.

The null-stimulus description indicated in the Results section needs further discussion and justification. This part should be rewritten extracting elements present in the methods. Also, the results are confusing. According to what is presented in Figure 5, the PSTH variance obtained from the white-noise response is larger than the variance of null-stimulus response, and at some extent, this variance would be reflecting inter-trial variability. At this level, this figure leads to confusion because it is not helping to understand the methodology. What is the role of the null-stimulus in this method paper? The method seems to predict the null-stimulus response better, but again, it is not clear how this clarifies the methodology neither provides insights to understand this behavior. It is counter-intuitive the fact that adding more units in the model subunits the response prediction for white-noise stimulus is not improving (Figure 6).

More detailed comments:

The manuscript was first submitted as original research before the submission as a method paper where essential changes in the structure and content were incorporated. Nevertheless, there is still information more related to retina physiology than the methodology description. Besides, if this is a method valid both for retina and V1 neurons, it is important to mention this in the title, where only the retina application is indicated. Therefore, please consider revising the article title.

In the Introduction, the authors claim there is no widely accepted general computation to infer the structure of nonlinear subunits inputs to a neuron. Nevertheless, relevant references have been published in the last years addressing the same problem noted by the authors. For instance, Liu et al., 2017, which appears in the discussion but not in the introduction, proposes a spike-triggered non-negative matrix factorization, to estimate the non-linear subunits conforming the RGC receptive field in salamander retina, where bipolar cells conforming RGCs can be also encountered.

It would be interesting to obtain and include some information about computational runtime.

The presentation of the analysis of the simulated ganglion cell under fine white noise stimulation (Figure 1—figure supplement 1C) is odd. Why are the subunits shown as sets of photoreceptors, not as pixel layouts? Does the method yield 12 subunits as the optimal number, or is this just the result of what came out when enforcing 12 subunits? A better discussion of how the method performs for simulated data could help better understand the strengths and limitations of the approach.

The figures show the "relative strength" of subunits by a blue scale bar. Is this the weight w? Or the norm of the subunit filter? Both seem to be related to "strength". And both seem to be optimized in the second step of the fitting approach along with the output nonlinearity.

Subsection “Estimated subunits are spatially localized and non-overlapping: The text refers to the Materials and methods section for explaining how the temporal filter was obtained, but the Materials and methods section seem to lack information about receptive field and temporal filter estimation.

Subsection “Regularization for spatially localized subunit estimation”: The chosen value of epsilon in the LNL1 regularization should somehow be related to the typical range of weights. Are the latter of order unity so that a value of epsilon=0.01 is small compared to typical weights?

Figure 4E: It seems odd that the negative log-likelihood keeps decreasing with the number of subunits in the joint fitting procedure. Unlike for the separate fitting. Is there a reason for this? Relatedly: how was the total number of subunits constrained for the separate fitting?

Subsection “Regularization for spatially localized subunit estimation”: The text states that a joint model with 15 subunits gives higher prediction accuracy, referring to Figure 4D. Does this relate to predictions as analyzed in Figure 5? Or to the maximum likelihood of Figure 4E?

Subsection “Subunit model explains spatial nonlinearities revealed by null stimulus”: The text refers to "spatial null spaces", but the Materials and methods section relates to spatiotemporal null spaces. Please clarify.

Subsection “Subunit model explains spatial nonlinearities revealed by null stimulus”, fourth paragraph: Maybe I missed it, but I wasn't sure which version of the subunit estimation is meant by "the subunit model". With or without regularization? Fine or coarse stimuli? A specific instance of the different initial conditions? Same for Figure 7.

Figure 7: The figure and the text around it use the term saccade, but the stimulus seems just to be a switch to a new image without a real motion segment between fixations. This is a bit confusing. Also, how the stimulus is segmented into periods around and between image switches should be better explained and not only mentioned in the legend, in particular how this is used to compute the projection on the receptive field (Figure 7C; is this a spatiotemporal projection?). Furthermore, the difference in Figure 7C seems quite subtle. Is it significant?

In the Materials and methods section, maybe state that the spatiotemporal stimulus was binary white noise (not Gaussian) if this is the case and that the X_t_ signal that goes into the subunit estimation is the contrast signal (zero mean).

Could the authors explain why the weights and the filter magnitudes need to be refitted in the second model fitting step?

Subsection “Application to neural responses”: What does the 5.2 µm boundary refer to?

For the simulated RGC model, the 180 µm (= 180 g.u.) spacing between cones seems large. Is that a typo? Also, in the description, it is not quite clear what the photoreceptor weights are (used for normalizing the firing rate) and how bipolar weights were varied around 10% of their mean (which mean?).

When describing the null stimulus, it would be good to explain where the original stimulus S_w_ comes from and how the σ for computing the support of the receptive field is obtained.

The references Maherwaranathan et al., 2018, and Shi et al., 2018 are missing.

Figures are in general of poor quality and the color choice misleading. In Figure 2D is not possible to see the black arrows over the blue bars. In Figure 2B blue line is not well visualized.

Figure 5D, use the same bar width for the inset histograms.

"… was highly reproducible across repeats (Figure 6C,D)," -> "… was highly reproducible across repeats (Figure 5C,D),"

Figure 6B: dashed are filled lines are not indicated in the caption.

Subsection "Application to neural response". The last three paragraphs should be incorporated into the Results section.

[Editors' note: further revisions were requested prior to acceptance, as described below.]

Thank you for resubmitting your work entitled "Inference of Nonlinear Receptive Field Subunits with Spike-Triggered Clustering" for further consideration at *eLife*. Your revised article has been favorably evaluated by Joshua Gold (Senior Editor), a Reviewing Editor, and three reviewers.

The manuscript has been improved but there are some remaining issues that need to be addressed before acceptance, as outlined below:

The comparison to the convolutional subunit model is not completely convincing for the following reasons: (1) The comparison done on V1 data uses a restricted version of the convolutional subunit model (the subunit nonlinearity is forced to be an exponential, despite that Vintch et al., (2015) and others have showed that it is often quadratic when learned from data). (2) No comparison is made on the retinal dataset where the fixed exponential subunit nonlinearity probably is a more realistic assumption. The reader is thus left wondering whether the new method truly has better predictive performance than the simpler convolutional subunit model, and whether the main benefit with the new method is that it can fit multiple subunit filters easily. It would be beneficial at least to clarify these concerns by giving a clearer justification to the selected methodology regarding the above issues.

A description for how/why the clustering algorithm's three step procedure solves the likelihood problem is still missing. It would be helpful if the authors could briefly spell out why the three-step prescription subsection “Subunit Estimation” “minimizes the upper bound of the log-likelihood (Equation 5). Are the update rules derived from the partial derivatives of Equation 5? Is this guaranteed to converge and converge to the actual minimum?

Regarding the validation in V1 data: More than a validation, where there is no ground-truth to compare, the results should be focused to validate the model showing that it reproduces neural data better than previous approaches. For instance, Figure 10 compares the resulting subunits obtained by a convolutional model and the model proposed in this article. It is tough to compare both; it should be more useful to see how the two of them reproduce neural response.

The Discussion section requires a new organization. The comparison with the existing models, such as SNMF and V1 data, shouldn't be placed here. The paragraph describing the work of McIntosh et al., 2016 can be better articulated with the rest of the Discussion section.

- For instance, the sentence "Note, however, that a model with two subunits frequently explained ON parasol data more accurately than a model with one subunit, implying that ON cells do have spatial nonlinearities". This is not related to the method validation, and moreover, there is no result to validate this affirmation.

- Subsection “Modeling linear-nonlinear cascades” states that the paper by Jia et al., (2018) applies a non-negativity constraint on the weights instead of the subunit filters. Looking at their methods and sample subunits in the figures, this doesn't seem to be the case.

- For the receptive field estimation in subsection “Application to neural responses”, the pixels with magnitude larger than 2.5 σ probably refer to magnitude **of the STA**; similarly, averaging the time course of pixels probably refers to averaging the pixels **in the STA**.

Figure 2 is still hard for me to follow. Is there a relationship between the grayscale images and the inner subunits? For instance, the subunits sizes go beyond the stimulus resolution, so, is their spatial location reliably estimated? Why level 3 only divides in two the left component, while the right one is still increasing the subunits number? In fact, the authors, in response to reviewers document and the article, indicate that the hierarchical partitioning is not used for the remainder in the paper. If this is not used, why is it introduced?

-Figure 5D. In the inset histograms, please use the same bin-width.

- Figure 6B. In the text, the authors talk about prediction accuracy. Nevertheless, the figure indicates Δ correlation. Same in Figure 7B.

---

## [Author Response]

The reviewers have discussed the reviews with one another and the Reviewing Editor has drafted this decision to help you prepare a revised submission.

We thank the reviewers for a thorough evaluation of the manuscript. In response to their feedback, we have modified the paper, and have provided responses to their feedback below.

Summary:[…]The main concern to address is that for a "tools and resources" papers needs to include clear benchmarking of the advantages of the current method against other existing and/or published approaches for subunit identification. The consensus among reviewers was that one would need to include comparison with at least one other method for both the retina and the V1 dataset. Some of the relevant methods that you could use are provided below, which offer different trade-offs between accuracy and ease of implementation and the amount of background work needed to set the hyperparameters. It would be helpful if the authors could include a discussion to justify their choice of the alternative method. The alternative method might be different for the V1 and the retina dataset. Including a more detailed discussion of the available methods and the authors' perspective on which would work best at each circumstance or generalize across the light/stimulus conditions would be very helpful to the field.The second concern was about the apparent inconsistencies in how the steps of the proposed algorithm were chosen (as detailed below) and that more methodological details need to be provided.

We have substantially improved the manuscript regarding both of these concerns. For the first, in addition to a detailed discussion of relationships to previous methods, we have included new analyses comparing the performance of our method to two others: the semi non-negative matrix factorization method (for retinal data), and the convolutional method (for V1 data). For the second concern, we have added an overview of the different algorithmic variations of the basic subunit estimation approach at the end of subsection “Subunit response model and parameter estimation”, and improved the text transitions at various places throughout the Results section to make the modifications for different experimental conditions clearer and better motivated. Details are provided below in response to the specific critiques.

Comparison with existing methods:While the manuscript includes some discussion of the existing methods, it is still not clear which model is the best taken into account that the subunits of the current model do not correlate with the actual bipolar cell receptive fields but rather with their clusters in some way. For the reader to take up this method, one would need to clearly see the pros and cons in speed, resolution, easiness of implementation, neural correlates of subunits and/or other key features of this model against other nonlinear subunit models (e.g. Liu et al., 2017 and the simpler earlier version by Kaardal et al., 2013) and/or simple convolution neural network models (Eickenberg et al., 2013, Vintch et al., 2015). It is critical for the reader to be able to assess the value of the current model. Assuming this is done properly, and these clarifications are added, this paper would be significantly improved.

We have added the following detailed comparisons:

Semi Non-negative Matrix Factorization (SNMF – similar to Liu et al., 2017): We have incorporated Figure 9 and an accompanying paragraph in the Discussion section comparing our method to SNMF in a simulated cell (for which ground truth is known). Briefly, we observed that our method is more efficient than SNMF at estimating subunits from limited data. Specifically, the fits using our method were more accurate and less variable.

A convolutional subunit model (similar to Vintch et al., 2015): We have incorporated Figure 10 and an accompanying paragraph in Discussion section comparing our method to an approach with convolutional (spatially-repeated, homogeneous) subunits. The convolutional method performs better for very small amounts of training data (as expected – it has far fewer parameters), but for larger amounts of data (comparable to what is available in typical experiments) our method performs better. The subunits estimated with our method roughly match both the locations and spatio-temporal selectivity of subunits estimated using the convolutional approach, but our method offers improved performance by capturing the distinct (non-convolutional) structure of individual subunits.

One would like to understand and document clearly performance of the subunit-identification method across stimulus conditions (white noise vs. naturalistic stimuli) against conventional currently used methods. This could perhaps be done by applying the new model to existing datasets of parasol RGCs in different experimental conditions. If such data are not available, then perhaps the method could be tested using simulated responses constructed to mimic properties of the retinal neurons.

If we understand correctly, the reviewer is asking us to (a) assess how well our method performs when applied to stimuli other than white noise (e.g., natural scenes), and (b) compare the performance of our method to the performance of previous methods (on at least two kinds of stimuli).

For (a), a direct comparison would involve obtaining and testing subunit model fits on natural scenes. We did not attempt to directly estimate subunits from natural scenes data, because natural scenes are biased toward large spatial scales, and are therefore ill-suited for estimating the fine structure of subunits in the RF. Indeed, even the linear estimate of the RF obtained using spike-triggered averaging is highly biased by natural scene structure. Because this problem is about the stimulus itself, it applies to all subunit estimation techniques. So, to test the effectiveness of the method for natural scenes, we instead estimated the subunits using white noise, then tested the success of these subunits in predicting the responses to natural scenes (Figure 7).

Regarding (b), we now provide a direct comparison of our method to the SNMF method (for RGCs) and the convolutional method (for V1), with white noise data. However, for technical reasons, we think comparison of natural scenes predictions would have little value. For both the SNMF approach and ours, extension to natural scenes data requires a modification to account for different response statistics in natural scenes. In both studies, this was handled with an ad hoc add-on procedure intended to demonstrate only that the subunits indeed help in response estimation (see Figure 7), but was *not* intended to fully capture responses to natural scenes (which likely entail much more complex nonlinearities than just subunits, e.g. various forms of gain control). We note that the authors of the SNMF study point this out explicitly. Our approach involves a modification of the output nonlinearity, and their approach involves an added response gain term, which are not easy to reconcile. In both studies, there is no exploration of alternatives or experimental test of the particular choice made. Furthermore, because the subunits themselves are estimated from white noise, a comparison of responses to natural scenes would hinge primarily on these ad hoc add-ons. For these reasons, we have not attempted this analysis -- regardless of the outcome, we don’t think it would do justice to either study.

Questions about the proposed methodology:There are some inconsistencies in the design of the proposed methodology. For examples, the method is motivated with an example that involves hierarchical splitting of subunits, but this hierarchical version is actually not used further on.

The basic method is not intended to be hierarchical. The hierarchical analysis is intended to illustrate (1) how the method behaves when the number of estimated subunits is less than the number of true subunits; (2) that the solutions for different numbers of estimated subunits are “nested”, in the sense that each successive increase in the number of subunits leads (approximately) to a splitting of a subunit into two. This observation is formalized in an analysis where hierarchy is enforced, confirming the result (Figure 2—figure supplement 1). This hierarchical partitioning can also be used as the basis for an incremental algorithm that provides a more robust/efficient method of estimating the subunits (Figure 2—figure supplement 1). Despite this, in the interest of simplicity, this is not the approach we use in the remainder of the paper. This has been clarified in the text. If the reviewer thinks this is too confusing, Figure 2—figure supplement 1 and accompanying analysis could be omitted.

Then the bare method (without regularization and fairly coarse stimuli) is used to analyze populations of OFF parasol cells (Figure 2B-F), but Figure 3 shows that much better subunits are obtained with regularization and finer white noise.

Early in the manuscript, we use the most basic form of the method (without priors/regularization) to establish its properties, without introducing any inadvertent biases or relying on any features of the particular data used. This resulted in subunits that were spatially localized, despite the fact that the method incorporates no preferences or knowledge regarding the spatial location of the stimulus pixels. This motivated the development of a prior that favors spatial locality (locally normalized L1), which we demonstrate can regularize RGC subunit estimates more efficiently than the more commonly used sparsity (L1) prior when data are limited. We have tried to clarify the motivation for this sequenced presentation in the text.

One wonders why this improved method wasn't used to obtain, for example, the distribution of subunit numbers.

As shown early in the manuscript, when the pixel size is large, the estimated subunits are aggregates of true underlying bipolar cell subunits. For the data with finer pixels, even though the estimated number of subunits is slightly higher, the pixel size and estimated subunits are still too large to reveal bipolar cell inputs. As such, re-analyzing the true number of subunits with regularization did not seem sufficiently informative given the space it would take.

Further analyses are then again (partly) performed without regularization (e.g. joint fitting of multiple cells, V1 data). In particular when discussing the V1 data, where the subunits have very different structure than in the retina, one wonders whether regularization still works in the same way.

Joint fitting of multiple cells was actually performed using the locally normalized L1 regularization that we showed to be effective for single cells. This was originally indicated only in the caption of Figure 4; We now also mention it in the main text and Methods section.

V1 subunits estimated without any regularization already revealed interesting structure in relation to previous findings, so we did not think it was necessary to incorporate any regularization (either the localization, or a new V1-specific variation). We have tried to be clearer about the rationale in the text.

Also, it is emphasized how the optimal number of subunits can be determined, but some of the subsequent analyses are performed for a fixed number of subunits (e.g. Figure 2 and Figure 3).

The appropriate approach for setting this parameter (the number of subunits) depends on what the analysis is being used for. The most straightforward case is when the user seeks to determine the number and structure of subunits for a particular cell. For this, we advocate cross-validated optimization -- choose the number of subunits that best describes the data (while not overfitting). This is empirically the “best” model fit, and is used in Figure 2C and Figure 8.

However, Figure 2A and Figure 3 have different goals:

- The goal of Figure 2A is to identify strong sources of spatial nonlinearity. By comparing the fits for different numbers of subunits, we find that the subunits are related hierarchically, suggesting that weaker subunits are partitions of stronger subunits.

- The goal of Figure 3 is to compare different regularization methods for subunit estimation. To avoid comparing different optimal number of subunits for each regularization method, the number of subunits are fixed to a value that works best across multiple cells.

We have tried to clarify the motivation for these analyses in the text.

There are also several other parts where that information is difficult to extract or missing, which hinders understanding of the material.The description of the subunit estimation approach via approximate maximum likelihood (subsection “Subunit estimation”) is central to the manuscript but quite difficult to follow. Some terms in the equation are not defined (K and w with superscript 0, c(K,w)), and the explanations for the derivations are cursory or absent. For example, when Equations 1-5 are derived, there is no mentioning of an iterative procedure (this only comes later; at this point, this seems to be only a reformulation of the log-likelihood), but some manipulations and terms here only make sense in the context of an iterative estimation procedure. This needs to be disentangled.Relatedly, it seems important to spell out how the clustering algorithm (Equations 6-8) solves the maximum likelihood problem, in particular since this connection is central to the manuscript.

We appreciate these comments and corrections, and have rewritten subsection “Subunit estimation” to make it more clear. Briefly, each step of the clustering algorithm results from minimizing a convex upper bound based on current parameter estimates. Equations 1-5 derive this upper bound and Equations 6-8 present the steps to minimize this upper bound. Omission of term definitions has been corrected.

The application of the method to V1 data makes an important point regarding the general applicability of the method. This is nice, but only very little analysis is presented. Maybe the authors could at least show how the model performs for different numbers of subunits as in other analyses, to see how well the simple cell and the complex cell are separated in this respect. Also, the effect of regularization would be interesting here.

We have incorporated the reviewer’s comments by adding analysis with varying number of subunits for V1 data, showing similar hierarchical splitting as RGCs (Figure 8).

Although the suggestion about simple and complex cells is an interesting one, given that the paper is now essentially a methods paper, and that such an analysis would require more space (as well as substantially more data), we have chosen not to pursue this direction.

Although in principle we could explore and test various regularization methods on V1 data, our goal with this analysis (and limited data set) was simply to demonstrate that the basic method works as well as or better than previous methods for subunit identification.

This assumption of space-time separability needs further discussion and justification. This assumption might be necessary to reduce the number of parameters to manageable levels, but it would be helpful if the authors could offer guidelines for when it might be appropriate.

We have added text in Subsection “Further applications and extensions” on how to verify the space-time separability assumption empirically. We have performed this verification on our data, but it would substantially lengthen the manuscript to include these data.

Note that the separability assumption is also made in other work that focuses on spatial subunits (Liu et al., 2017) and implicitly or explicitly in much of the literature on RGCs (but see Enroth-Cugell and Pinto., 1970 for a deeper investigation of separability). Note also that separability is not required by our subunit model or fitting procedure -- for example, in the V1 cells, the method extracts spatio-temporally oriented (and thus, non-separable) receptive fields.

About the response prediction to naturalistic stimuli. The use of white noise stimuli rather than natural stimuli might provide a good approximation for estimating of subunit filters, However, bipolar cells, which provide inputs to parasol RGCs, are strongly affected by stimulus correlation through gap-junction connectivity generating a nonlinear enhancement of bipolar activity, which could explain the differences found with real activity (Kuo et al., 2016, Manookin et al., 2018). These elements should be included in the Discussion section as one of the potential pitfalls of the approach.

Thank you for pointing this out. We now explain the possible role of gap junctions in natural scenes responses in Results section and Discussion section.

The null-stimulus description indicated in the Results section needs further discussion and justification. This part should be rewritten extracting elements present in the methods.

We have rewritten this section with more specifics and detail as suggested.

Also, the results are confusing. According to what is presented in Figure 5, the PSTH variance obtained from the white-noise response is larger than the variance of null-stimulus response, and at some extent, this variance would be reflecting inter-trial variability.

We thank the reviewer for pointing this out. Indeed, the PSTH variance we measure has a component of response variation over time (the thing we are interested in) and inter-trial variability (which is of less interest here). On evaluating the effect of the number of trials, we find that PSTH variance converges with 30 trials, in 3 datasets (Author response image 1); thus, the inter-trial variability is not significant in the analysis performed in the paper. We have now indicated this in Results.

**Author response image 1. respfig1:** Convergence of PSTH variance. (**A**) PSTH variance (y-axis), averaged over ON and OFF populations for white noise (black) and null stimulus (red) as a function of the number of randomly sampled trials (x-axis). Line thickness corresponds to +/- 1 s.d. in estimation error. Same retina as in Figure 5, main paper. (**B, C**) Same as A, for two other retinas.

At this level, this figure leads to confusion because it is not helping to understand the methodology. What is the role of the null-stimulus in this method paper? The method seems to predict the null-stimulus response better, but again, it is not clear how this clarifies the methodology neither provides insights to understand this behavior. It is counter-intuitive the fact that adding more units in the model subunits the response prediction for white-noise stimulus is not improving (Figure 6).

Null stimuli provide a direct means of demonstrating the importance of the subunit nonlinearity. They expose spatial nonlinearities by specifically “zeroing out” the component of light response that can be explained with a linear model. In this sense, they are analogous to the commonly-used counterphase gratings and second harmonic analysis (Hochstein and Shapley, 1976), which have been used in dozens of studies to focus on bipolar cell nonlinearities. The null stimuli, however, have much richer stimulus statistics (contrast, temporal frequency, spatial scale) very similar to the white noise stimuli that are used for estimating the subunit model that is being tested. The counter-intuitive result that the reviewer points out is that the efficiency of the null stimuli at revealing spatial nonlinearities allows them to reveal a clear improvement of subunit models even for short duration stimuli (Figure 6), while white noise stimuli require longer stimulus presentations (Figure 2).

We have attempted to clarify these points in the text.

More detailed comments:The manuscript was first submitted as original research before the submission as a method paper where essential changes in the structure and content were incorporated. Nevertheless, there is still information more related to retina physiology than the methodology description. Besides, if this is a method valid both for retina and V1 neurons, it is important to mention this in the title, where only the retina application is indicated. Therefore, please consider revising the article title.

We agree, and have revised the title accordingly, – “Inference of Nonlinear Receptive Field Subunits with Spike-Triggered Clustering”

In the Introduction, the authors claim there is no widely accepted general computation to infer the structure of nonlinear subunits inputs to a neuron. Nevertheless, relevant references have been published in the last years addressing the same problem noted by the authors. For instance, Liu et al., 2017, which appears in the discussion but not in the introduction, proposes a spike-triggered non-negative matrix factorization, to estimate the non-linear subunits conforming the RGC receptive field in salamander retina, where bipolar cells conforming RGCs can be also encountered.

We agree that the recent literature on this topic is active: various groups have realized that subunits are a common computational motif, and that new tools are required to study their role in neural computation. We have revised the introduction to position our contribution relative to these studies (although we note that the Liu et al., 2017 study was already mentioned in paragraph 2 of our original Introduction). In particular, although previous efforts to identify subunits have been advanced in various experimental conditions, none of the existing papers have developed a single method and applied it to multiple stimulus conditions, with different statistics, different systems, and large numbers of recorded neurons.

It would be interesting to obtain and include some information about computational runtime.

Thanks for this suggestion – we have added information about computational runtime in Materials and methods section.

The presentation of the analysis of the simulated ganglion cell under fine white noise stimulation (Figure 1—figure supplement 1C) is odd. Why are the subunits shown as sets of photoreceptors, not as pixel layouts? Does the method yield 12 subunits as the optimal number, or is this just the result of what came out when enforcing 12 subunits? A better discussion of how the method performs for simulated data could help better understand the strengths and limitations of the approach.

Unfortunately, the text in the caption was confusing. While the coarse pixel approach is described in panels A and B, in panel C we simulate what would happen if each cone were driven independently. Although this would require a closed-loop experiment identifying individual cone locations (Freeman, 2015), we show here that it would reveal the correct subunits with the estimation approach in this paper. We have attempted to explain the simulation better in the caption (including indication that the correct number of subunits (12) was identified), and motivate this choice of representation better in the text.

The figures show the "relative strength" of subunits by a blue scale bar. Is this the weight w? Or the norm of the subunit filter? Both seem to be related to "strength". And both seem to be optimized in the second step of the fitting approach along with the output nonlinearity.

The blue bars correspond to the average contribution to cell response for different subunits given by: wne||Kn||22. We now provide a pointer to Equation (6) in the methods. This formula ignores the learned output non-linearity, which saturates only for the strongest 0.5% of the stimuli. This information has been added in Results section and Materials and methods section.

Subsection “Estimated subunits are spatially localized and non-overlapping: The text refers to the Materials and methods section for explaining how the temporal filter was obtained, but the Materials and methods section seem to lack information about receptive field and temporal filter estimation.

Thank you for spotting this. We’ve added the explanation to the Materials and methods section.

Subsection “Regularization for spatially localized subunit estimation”: The chosen value of epsilon in the LNL1 regularization should somehow be related to the typical range of weights. Are the latter of order unity so that a value of epsilon=0.01 is small compared to typical weights?

Yes, typically, the nonzero weights were ~2 and the maximum weights were ~4. This information has been added to the paper.

Figure 4E: It seems odd that the negative log-likelihood keeps decreasing with the number of subunits in the joint fitting procedure. Unlike for the separate fitting. Is there a reason for this?

Unlike other figures, the negative log-likelihood for the joint model is decreasing for the range of subunits plotted in Figure 4E. This can be attributed to the greater efficiency of joint estimation procedure compared to separate fitting. For separate fitting, a subunit connected to two nearby cells needs to be estimated by both the cells independently.

We believe that the negative log-likelihood for the joint model will indeed saturate, but for larger number of subunits than currently shown. For presenting a concise figure, we did not analyze the model for larger number of subunits.

Relatedly: how was the total number of subunits constrained for the separate fitting?

First, for each cell, models with different number of subunits are estimated. Then, for a fixed total number of subunits, these separate fits are used to find a combination of subunits (*N_i_* for cell i) that gives maximum log-likelihood on held-out validation data and ∑Ni=N. Performance is analyzed for different *N*. This has been clarified in the main text.

Subsection “Regularization for spatially localized subunit estimation”: The text states that a joint model with 15 subunits gives higher prediction accuracy, referring to Figure 4D. Does this relate to predictions as analyzed in Figure 5? Or to the maximum likelihood of Figure 4E?

Thank you for pointing out this typo. It refers to Figure 4E.

Subsection “Subunit model explains spatial nonlinearities revealed by null stimulus”: The text refers to "spatial null spaces", but the Materials and methods section relates to spatiotemporal null spaces. Please clarify.

The methods section presents the steps involved in a more general, spatio-temporal null stimulus, as well as a method for spatial-only null stimulus, and the relationship between them. In case of RGCs, where the STA is space-time separable, the spatial-only nulling is equivalent to spatio-temporal nulling. However, for the V1 data, the more general spatio-temporal nulling method needs to be applied, and this is why we have included it in the Materials and methods section.

Subsection “Subunit model explains spatial nonlinearities revealed by null stimulus”, fourth paragraph: Maybe I missed it, but I wasn't sure which version of the subunit estimation is meant by "the subunit model". With or without regularization? Fine or coarse stimuli? A specific instance of the different initial conditions? Same for Figure 7.

For both Figure 6 and Figure 7, the unregularized method was applied on coarse stimuli; the first phase of fitting was applied on white noise and second phase was applied on null stimulus or natural scenes respectively. This has now been clarified in the text.

Figure 7: The figure and the text around it use the term saccade, but the stimulus seems just to be a switch to a new image without a real motion segment between fixations. This is a bit confusing.

This has been clarified in the main text. We have emphasized that this is only a crude approximation of visual input interrupted by saccades.

Also, how the stimulus is segmented into periods around and between image switches should be better explained and not only mentioned in the legend, in particular how this is used to compute the projection on the receptive field (Figure 7C; is this a spatiotemporal projection?).

Yes, it is spatio-temporal filtering. This has been clarified in the main text.

Furthermore, the difference in Figure 7C seems quite subtle. Is it significant?

We agree that the difference in the histograms seems subtle – one has to look for a moment to see that the black bars are lower in the center, and higher at the extremes. However, the two distributions are significantly different as evaluated by comparing their variances using a two-sided F-test. The variances for saccadic and inter-saccadic stimuli were 233 (N=5e6) and 155 (N=15e6), respectively.

In the Materials and methods section, maybe state that the spatiotemporal stimulus was binary white noise (not Gaussian) if this is the case and that the X_t_ signal that goes into the subunit estimation is the contrast signal (zero mean).

This has been added.

Could the authors explain why the weights and the filter magnitudes need to be refitted in the second model fitting step?

Since the output nonlinearity changes from linear to a more saturating one in natural scenes, the overall firing rate of the model cell is lower, and the weights on subunits and/or their magnitudes must be increased to match the observed firing rate. Moreover, the saturating nonlinearity could change the relative weights for different subunits – subunits that drive the cell more strongly would be more affected by saturation. However, on applying to data, we observed that the final nonlinearity is nearly linear, confirming that the first model fitting step is sufficient in most cases.

Subsection “Application to neural responses”: What does the 5.2 µm boundary refer to?

This refers to the region of stimulus around the receptive field which is included for subunit estimation. This has been clarified in the Materials and methods section.

For the simulated RGC model, the 180 µm (= 180 g.u.) spacing between cones seems large. Is that a typo? Also, in the description, it is not quite clear what the photoreceptor weights are (used for normalizing the firing rate) and how bipolar weights were varied around 10% of their mean (which mean?).

It is a typo – thanks for pointing it out. The spacing between photoreceptors is 5 µms. The details of photoreceptor and bipolar weights have been corrected. The bipolar weights were drawn from a gaussian distribution with standard deviation equal to 10% of the mean.

When describing the null stimulus, it would be good to explain where the original stimulus S_w_ comes from and how the σ for computing the support of the receptive field is obtained.

The original stimulus S*_w_* is white noise used for computing the null stimulus and σ is a robust estimate of the standard deviation of background noise in STA. We have clarified in the text.

The references Maherwaranathan et al., 2018, and Shi et al., 2018 are missing.

These recent references have been added.

Figures are in general of poor quality and the color choice misleading.

We apologize for the low-resolution figures. They have been replaced with high resolution versions. We chose to focus on as few colors as possible and the color choices are consistent across figures. The color choices for white noise and null stimulus in Figure 5 and Figure 6 were interchanged. This is fixed now.

In Figure 2D is not possible to see the black arrows over the blue bars. In Figure 2B blue line is not well visualized.

For Figure 2D, we moved the black arrows to the top for clarity. The blue line is made continuous (instead of dotted) for better visualization.

Figure 5D, use the same bar width for the inset histograms.

We agree that having different bin widths makes the figure visually awkward. However, since the range of values for ON and OFF parasols is different, a common bin-width does not provide a good depiction of both distributions. Instead, we chose the same number of bins to describe both distributions equally accurately.

"… was highly reproducible across repeats (Figure 6C,D)," -> "… was highly reproducible across repeats (Figure 5C,D),"

This has been fixed.

Figure 6B: dashed are filled lines are not indicated in the caption.

We had mistakenly referred to them as ‘dotted’ lines in caption. Now changed to ‘dashed’ lines.

Subsection "Application to neural response". The last three paragraphs should be incorporated into the Results section.

The information from these paragraphs is now included in the main text in Results section and/or captions.

[Editors' note: further revisions were requested prior to acceptance, as described below.]

The manuscript has been improved but there are some remaining issues that need to be addressed before acceptance, as outlined below:

Thank you for your favourable review. Below, we address the remaining issues.

The comparison to the convolutional subunit model is not completely convincing for the following reasons: (1) The comparison done on V1 data uses a restricted version of the convolutional subunit model (the subunit nonlinearity is forced to be an exponential, despite that Vintch et al., (2015) and others have showed that it is often quadratic when learned from data). (2) No comparison is made on the retinal dataset where the fixed exponential subunit nonlinearity probably is a more realistic assumption. The reader is thus left wondering whether the new method truly has better predictive performance than the simpler convolutional subunit model, and whether the main benefit with the new method is that it can fit multiple subunit filters easily. It would be beneficial at least to clarify these concerns by giving a clearer justification to the selected methodology regarding the above issues.

(1) We agree with the reviewers, and have now added a relevant comparison (Figure 10—figure supplement 1). For the convolutional model, we estimate the common subunit filter, location-specific weights and biases for quadratic nonlinearity using standard gradient descent methods. For our model, we first estimate the subunit filters with spike-triggered clustering (which assumes an exponential nonlinearity), and fine-tune the subunits filters and bias for quadratic nonlinearity using gradient descent. We find that both methods perform better with a quadratic nonlinearity, consistent with previous studies (Vintch et al., 2015). But as with the exponential nonlinearity, the convolutional method outperforms the spike-triggered clustering method when trained on small amounts of data, but the opposite holds for larger amounts of data. (2) As requested, we have added a comparison on the retinal data, which produces similar results to the V1 data (Figure 10).

A description for how/why the clustering algorithm's three step procedure solves the likelihood problem is still missing. It would be helpful if the authors could briefly spell out why the three-step prescription subsection “Subunit estimation” “minimizes the upper bound of the log-likelihood (Equation 5). Are the update rules derived from the partial derivatives of Equation 5? Is this guaranteed to converge and converge to the actual minimum?

We have expanded the description and derivation of the algorithm in Methods. We now explain how the update rules are derived from partial derivatives of Equation 5. In brief: we are minimizing an approximation to the negative log-likelihood (Equation 2). As the amount of data grows, the approximation converges to the negative log-likelihood. The algorithm is guaranteed to converge since it successively minimizes a tight upper bound to the approximate log-likelihood, leading to a monotonic increase in the approximate log-likelihood. However, due to the non-convexity of the approximate negative log-likelihood, we cannot guarantee that the point of convergence will be a global optimum.

Regarding the validation in V1 data: More than a validation, where there is no ground-truth to compare, the results should be focused to validate the model showing that it reproduces neural data better than previous approaches. For instance, Figure 10 compares the resulting subunits obtained by a convolutional model and the model proposed in this article. It is tough to compare both; it should be more useful to see how the two of them reproduce neural response.

We agree with the reviewer that in addition to the qualitative comparison of the derived filters, it is valuable to compare predictions of neural response. This is done by plotting the log-likelihood of the recorded response as a function of the amount of training data in Figure 10C (and in the new Figure 10A as well for retinal data). We have attempted to clarify this in the figure caption.

The Discussion section requires a new organization. The comparison with the existing models, such as SNMF and V1 data, shouldn't be placed here. The paragraph describing the work of McIntosh et al., 2016 can be better articulated with the rest of the Discussion section.

As suggested, we have moved the comparison to SNMF and convolutional models to Results section and reorganized the relevant section of Discussion section.

- For instance, the sentence "Note, however, that a model with two subunits frequently explained ON parasol data more accurately than a model with one subunit, implying that ON cells do have spatial nonlinearities". This is not related to the method validation, and moreover, there is no result to validate this affirmation.

This statement was intended to make the point that, even though the optimum number of subunits for ON parasol cells was lower than for OFF parasol cells, spatial nonlinearity was still present in ON parasol cells. To support this claim, we have now provided statistics in the text, and rephrased it slightly.

- Subsection “Modeling linear-nonlinear cascades” states that the paper by Jia et al., (2018) applies a non-negativity constraint on the weights instead of the subunit filters. Looking at their methods and sample subunits in the figures, this doesn't seem to be the case.

The reviewer is absolutely correct: we made a mistake in our interpretation of the 2018 paper. It has now been corrected in the manuscript.

- For the receptive field estimation in subsection “Application to neural responses”, the pixels with magnitude larger than 2.5 σ probably refer to magnitude **of the STA**; similarly, averaging the time course of pixels probably refers to averaging the pixels **in the STA**.

Thanks for pointing this out. It has been corrected.

Figure 2 is still hard for me to follow. Is there a relationship between the grayscale images and the inner subunits? For instance, the subunits sizes go beyond the stimulus resolution, so, is their spatial location reliably estimated? Why level 3 only divides in two the left component, while the right one is still increasing the subunits number? In fact, the authors, in response to reviewers document and the article, indicate that the hierarchical partitioning is not used for the remainder in the paper. If this is not used, why is it introduced?

We apologize for lack of clarity in this figure. We suspect the confusion is due to a poor graphics choice on our part that made it difficult for a reader to see the Gaussian fits to the subunits. We have modified the figure so that the Gaussian fit for the fitted subunit shown in the image is indicated with a green ellipse that is more visible on darker pixels, and used red ellipses to represent the other estimated subunits. We hope that this makes the partitioning of receptive field clearer.

As the reviewer points out, the subunit estimates are limited by stimulus pixel resolution. For this reason, we measure the separation between subunits using the peaks of their Gaussian fits which, assuming that the true subunits are roughly Gaussian, effectively interpolates their center locations.

Regarding the last comment, we thought it was important to mention that enforcing hierarchical partitioning (Figure 2—figure supplement 1) leads to similar subunits, because this reinforces the idea that subunit estimates effectively “split” as the number of subunits in the model increases. Also, enforcing this hierarchical organization could lead to improved estimates by providing a form of regularization. We have further clarified this in the text. If the reviewers still think that these points are not important, we could remove ‘Figure 2—figure supplement 1’.

-Figure 5D. In the inset histograms, please use the same bin-width.

Done.

- Figure 6B. In the text, the authors talk about prediction accuracy. Nevertheless, the figure indicates Δ correlation. Same in Figure 7B.

We apologize for the confusion about the metric. Throughout the paper, we use negative log-likelihood for measuring accuracy for single-trial data, and correlation for raster data. For studying the variation of performance with increasing numbers of subunits, the *change* in correlation compared to one subunit is presented in the figures, because there is a comparatively large variation across cells in the correlation obtained with one subunit, which would obscure the trends seen in the figure. We have modified the figure caption to make this connection clearer.